# Disentangling and Integrating Relational and Sensory Information in Transformer Architectures

**Awni Altabaa** [1]   **John Lafferty** [2]

## Abstract

Relational reasoning is a central component of generally intelligent systems, enabling robust and data-efficient inductive generalization. Recent empirical evidence shows that many existing neural architectures, including Transformers, struggle with tasks requiring relational reasoning. In this work, we distinguish between two types of information: *sensory* information about the properties of individual objects, and *relational* information about the relationships between objects. While neural attention provides a powerful mechanism for controlling the flow of sensory information between objects, the Transformer lacks an explicit computational mechanism for routing and processing relational information. To address this limitation, we propose an architectural extension of the Transformer framework that we call the *Dual Attention Transformer (DAT)*, featuring two distinct attention mechanisms: sensory attention for directing the flow of sensory information, and a novel relational attention mechanism for directing the flow of relational information. We empirically evaluate *DAT* on a diverse set of tasks ranging from synthetic relational benchmarks to complex real-world tasks such as language modeling and visual processing. Our results demonstrate that integrating explicit relational computational mechanisms into the Transformer architecture leads to significant performance gains in terms of data efficiency and parameter efficiency.

## 1. Introduction

A central goal of machine learning research is to develop a universal architecture capable of learning and reasoning across a wide range of tasks and data modalities. Theoretical frameworks for understanding human and animal intelligence seek to explain intelligent behavior through a small set of fundamental principles [1]. However, in machine intelligence, there is a tension between the objective of developing a *general* architecture and the need to incorporate *inductive biases* that are beneficial for specific tasks [2, 3]. When faced with finite training data and numerous solutions to empirical risk minimization, inductive biases steer the learning algorithm towards solutions with desirable properties, enhancing data efficiency and generalization. A core scientific challenge of machine learning is to identify a complete and broadly applicable set of inductive biases that promote robust, flexible, and data-efficient learning across a diverse set of problems.

Relational reasoning is a central component of generally intelligent systems and is believed to underlie human abilities for abstraction and systematic generalization [4–7]. The power of relational reasoning lies in its capacity to generate inferences and generalizations in systematic and novel ways, even across instances which are superficially very different but have similarities on an abstract structural level. This grants humans a capacity for out-of-distribution generalization, data efficiency, and continual learning, which modern machine learning is yet to match [8–10]. Replicating this ability in machine intelligence can ultimately lead to universal inductive generalization from a finite set of observations to an infinite set of novel instances [11].

The ability of artificial intelligence systems to perform relational reasoning has long been an area of active study across various approaches to AI. In symbolic modeling frameworks, the relationships between symbols are explicitly defined in the language of logic and mathematics [12–14]. However, such approaches often require hand-crafted representations and suffer from the symbol grounding problem, limiting their ability to generalize to variations in the task or input outside a certain domain. By contrast, deep learning approaches build data-dependent representations that are, in principle, capable of generalizing across diverse conditions.

[1]Department of Statistics and Data Science, Yale University [2]Department of Statistics and Data Science, Wu Tsai Institute, Yale University. Correspondence to: Awni Altabaa <awni.altabaa@yale.edu>, John Lafferty <john.lafferty@yale.edu>.

*Proceedings of the 42nd International Conference on Machine Learning*, Vancouver, Canada. PMLR 267, 2025. Copyright 2025 by the author(s).

However, recent work exploring the ability of deep learning models to learn relational tasks finds that seemingly simple relational inferences can be remarkably difficult for powerful neural network architectures [15–23]. An emerging hypothesis, which we explore further here, explains this through the lens of inductive biases, arguing that neural networks struggle with relational reasoning because they overemphasize individual object representations—*sensory* information—while lacking *explicit* mechanisms for encoding and processing *relational information* [24].

In this work, we explore this idea in the context of the Transformer architecture [25], which offers a promising starting point for building a versatile, general-purpose neural architecture. Although Transformers, like other neural architectures, struggle to learn abstract relational representations [19–23], there is encouraging evidence that Transformer-based foundation models acquire some relational reasoning ability [26–32] through large-scale training on large amounts of data [33–36]. This presents an opportunity to explore Transformer-based architectures with built-in neural mechanisms and inductive biases for explicit and enhanced relational reasoning capabilities, seeking to imbue the Transformer architecture with a greater capacity for efficient induction of abstractions.

To introduce our proposal, we highlight a distinction between two types of information which are encoded in the internal representation of Transformer models: *sensory information*, which represents features of individual objects, and *relational information*, representing comparisons and relationships between objects. In the standard attention mechanism of Transformers, relational information is *entangled* with sensory information, limiting the model's ability to learn to explicitly represent and reason about the relationships between objects. In particular, in standard attention, relational information *directs* the flow of information (i.e., attention scores encoding a selection criterion), while the *values* routed are representations of the sensory attributes of individual objects. In this work, we explore a new type of attention mechanism we call *relational attention*, in which the values being routed are themselves representations of relationships between the source and the target, computed explicitly as a series of comparisons under learned feature maps. This equips the model with a mechanism for explicitly routing and processing relational information, decoupling it from sensory features.

We integrate relational attention with the standard attention mechanism of Transformers, yielding a variant of multi-head attention that processes both sensory and relational information in parallel. The resulting *Dual Attention Transformer (DAT)* architecture disentangles the two types of information during the information retrieval stage and integrates them during the local processing stage of each layer.

We empirically evaluate this architecture on a diverse set of tasks, ranging from synthetic benchmarks on relational reasoning to complex real-world tasks such as language modeling and visual processing. Our results demonstrate that integrating explicit relational mechanisms into the Transformer architecture leads to significant performance gains in terms of data efficiency and parameter efficiency.

## 2. Disentangling Attention over Sensory and Relational Information

### 2.1. Standard Attention: Attention over Sensory Information

The attention mechanism of standard Transformers can be understood as a differentiable computational mechanism for dynamically routing sensory information between different elements in the input. An object emits a query that is compared against the keys of each object in its context via an inner product. A "match" occurs when the inner product is large, causing an encoding of the features of the attended object to be retrieved and added to the residual stream of the receiver. Formally, attention between an object $x \in \mathbb{R}^d$ and a context $\boldsymbol{y} = (y_1, \ldots, y_n) \in \mathbb{R}^{n \times d}$ takes the form

$$\text{Attention}(x, (y_1, \ldots, y_n))$$
$$= \sum_{i=1}^{n} \alpha_i(x, \boldsymbol{y}) \phi_v(y_i), \text{ where,} \quad (1)$$
$$\alpha(x, \boldsymbol{y}) = \text{Softmax}\big( \big[ \langle \phi_q^{\text{attn}}(x), \phi_k^{\text{attn}}(y_i) \rangle \big]_{i=1}^{n} \big),$$

where $\phi_q^{\text{attn}}, \phi_k^{\text{attn}}$ are learnable query and key maps controlling the selection criterion, and $\phi_v$ is a learnable value map controlling what information about $y_i$ is sent. The attention scores $\alpha(x, \boldsymbol{y})$ are used to retrieve a convex combination of the values, where $\alpha_i(x, \boldsymbol{y})$ denotes the $i$-th component.

Here, the retrieved information is *sensory*, comprising the features and attributes of individual objects in the context. For this reason, we refer to standard neural attention as "sensory attention".

### 2.2. Relational Attention: Attention over Relational Information

Standard neural attention does not explicitly capture information about the *relationship* between the source/sender and the target/receiver, making relational learning in standard Transformers inefficient [15, 16, 18–24, 37]. We propose *relational attention*, an attention mechanism for dynamically routing relational information between objects.

Mirroring standard attention, this operation begins with each object emitting a query and a key, which are compared via an inner product to compute attention scores determining which objects to attend to. Next, instead of retrieving the

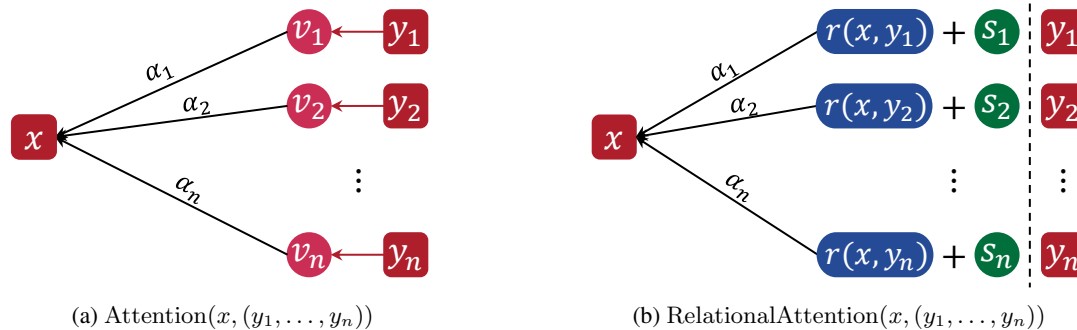

(a) $\text{Attention}(x, (y_1, \ldots, y_n))$      (b) $\text{RelationalAttention}(x, (y_1, \ldots, y_n))$

*Figure 1.* Standard self-attention retrieves sensory information $v_i$ about the attributes of individual objects while relational attention retrieves relational information $r(x, y_i)$ about the relationship between the objects in the context and the target. Each relation is tagged with a symbol $s_i$ which acts as an abstract variable identifying the source. In both cases, information is aggregated according to the attention scores $\alpha_i$, which are computed by a softmax over inner products of queries and keys.

sensory features of the selected object, relational attention retrieves the *relation* between the two objects—defined as a series of comparisons between the two objects under different feature subspaces. In addition, a symbolic identifier is sent to indicate the identity of the sender to the receiver. Formally, this operation is defined as follows.

$$\text{RelationalAttention}(x, (y_1, \ldots, y_n))$$
$$= \sum_{i=1}^{n} \alpha_i(x, \boldsymbol{y})\big(r(x, y_i)W_r + s_i W_s\big), \text{ where,}$$
$$\alpha(x, \boldsymbol{y}) = \text{Softmax}\big(\,\big[\langle \phi_q^{\text{attn}}(x), \phi_k^{\text{attn}}(y_i)\rangle\big]_{i=1}^{n}\,\big),$$
$$r(x, y_i) = \big(\langle \phi_{q,\ell}^{\text{rel}}(x), \phi_{k,\ell}^{\text{rel}}(y_i)\rangle\big)_{\ell \in [d_r]},$$
$$(s_1, \ldots, s_n) = \text{SymbolRetriever}(\boldsymbol{y}; S_{\text{lib}})$$
$$(2)$$

Thus, relational attention between the object $x$ and the context $\boldsymbol{y} = (y_1, \ldots, y_n)$ retrieves a convex combination of the relation vectors $\{r(x, y_i)\}_{i=1}^{n}$, representing $x$'s relationship with each object in the context. Relational attention also retrieves a symbol vector $s_i$ that encodes the identity information of the attended object. The role and implementation of the symbols will be discussed in the next subsection. As with standard attention, $\phi_q^{\text{attn}}, \phi_k^{\text{attn}}$ are learned feature maps that govern the attention selection criterion. A separate set of query and key feature maps, $\phi_{q,\ell}^{\text{rel}}, \phi_{k,\ell}^{\text{rel}}, \ell \in [d_r]$, are learned to represent the relation between the sender and the receiver. For each $\ell \in [d_r]$, the feature maps $\phi_{q,\ell}^{\text{rel}}, \phi_{k,\ell}^{\text{rel}}$ extract specific attributes from the object pair, which are compared by an inner product. This produces a $d_r$-dimensional relation vector representing a fine-grained series of comparisons $(\langle \phi_{q,\ell}^{\text{rel}}(x), \phi_{k,\ell}^{\text{rel}}(y_i)\rangle)_{\ell \in [d_r]}$ across different feature subspaces.

In certain tasks [21–23], a useful inductive bias on the relations function $r(\cdot, \cdot)$ is symmetry; i.e., $r(x, y) = r(y, x), \forall x, y$. This corresponds to using the same feature filter for the query and key maps, $\phi_q^{\text{rel}} = \phi_k^{\text{rel}}$. This adds

structure to the relation function, transforming it into a positive semi-definite kernel that defines a pseudometric on the object space, along with a corresponding geometry.

### 2.3. Symbol Assignment Mechanisms

To process relational information effectively, the receiver must have two pieces of information: 1) its relationship to the objects in its context, and 2) the identity of the object associated with each relation. In relational attention, the former is captured by $r(x, y_i)$ and the latter by $s_i$. The symbols $s_i$ are used to tag each relation with the identity information of the sender.

The symbol $s_i$ identifies or points to the object $y_i$, but, importantly, is designed to not fully encode the features of the object. Instead, the symbols $s_i$ function as abstract references to objects, perhaps viewed as a connectionist analog of pointers in traditional symbolic systems. In particular, by drawing symbol vectors from a finite library $S_{\text{lib}}$, relational attention maintains a relation-centric representation. This separation between sensory and relational information is key to decoupling relational attention disentangled from sensory features, enabling generalization across relations.

The notion of the "identity" of an object can vary depending on context. In this work, we consider modeling three types of identifiers: 1) position, 2) relative position, or 3) an equivalence class over features. For each type of identifier, we model a corresponding symbol assignment mechanism [22]. We find that different symbol assignment mechanisms are more effective in different domains.

**Positional Symbols.** In some applications, it is sufficient to identify objects through their position in the input sequence. We maintain a library of symbols $S_{\text{lib}} = (s_1, \ldots, s_{\texttt{max\_len}}) \in \mathbb{R}^{\texttt{max\_len} \times d}$ and assign $s_i$ to the $i$-th object in the sequence. These are essentially learned positional embeddings.

**Position-Relative Symbols.** Often, the *relative* position with respect to the receiver is a more useful identifier than absolute position. This can be implemented with position-relative embeddings. We learn a symbol library $S_{\text{lib}} = (s_{-\Delta}, \ldots, s_{-1}, s_0, s_1, \ldots, s_\Delta) \in \mathbb{R}^{(2\Delta+1)\times d}$, where $\Delta$ is the maximum relative position, and relational attention becomes $\sum_j \alpha_{ij}(r(x_i, x_j) W_r + s_{j-i} W_s)$.

**Symbolic Attention.** In certain domains, some information about the objects' features is necessary to identify them for the purposes of relational processing. Yet, to maintain a relational inductive bias, we would like to avoid a *full* encoding of object-level features. In symbolic attention, we learn a set of symbol vectors, $S_{\text{lib}} = (s_1, \ldots, s_{n_s}) \in \mathbb{R}^{n_s \times d}$ and a matching set of feature templates $F_{\text{lib}} = (f_1, \ldots, f_{n_s})$. We retrieve a symbol for each object by an attention operation that matches the input vectors $x_i$ against the feature templates $f_j$ and retrieves symbols $s_j$.

$$\text{SymbolicAttention}(\boldsymbol{x}) = \text{Softmax}\big((\boldsymbol{x} W_q) F_{\text{lib}}^\top\big) S_{\text{lib}}. \tag{3}$$

Here, $S_{\text{lib}}, F_{\text{lib}}, W_q$ are learned parameters. This can be thought of as implementing a learned differentiable "equivalence class map" over feature embeddings. Crucially, the number of symbols (i.e., feature equivalence classes) is *finite*, which enables relational attention to still produce a relation-centric representation while tagging the relations with the necessary identifier.

### 2.4. What Class of Functions can Relational Attention Compute?

To give some intuition about the type of computation that relational attention can perform, we present the following expressivity result. The following theorem states that relational attention can approximate any function on $\mathcal{X} \times \mathcal{Y}^n$ that 1) selects an element in $(y_1, \ldots, y_n)$, then 2) computes a relation with it. Both the selection criterion and the relation function are arbitrary, and the selection criterion can be query-dependent. The formal statement and proof are given in Appendix A.

**Theorem 1** (Informal). *Let* $\text{Select} : \mathcal{X} \times \mathcal{Y}^n \to \mathcal{Y}$ *be an arbitrary preference selection function, which selects an element among* $(y_1, \ldots, y_n)$ *based on a query-dependent preorder relation* $\{\preccurlyeq_x\}_{x \in \mathcal{X}}$. *Let* $\text{Rel} : \mathcal{X} \times \mathcal{Y} \to \mathbb{R}^{d_r}$ *be an arbitrary continuous relation function on* $\mathcal{X} \times \mathcal{Y}$. *There exists a relational attention module that approximates the function* $\text{Rel}(x, \text{Select}(x, \boldsymbol{y}))$ *to arbitrary precision.*

---

**Algorithm 1:** Dual Attention

**Input:** $\boldsymbol{x} = (x_1, \ldots, x_n) \in \mathbb{R}^{n \times d}$

```
Compute self-attention heads
```
$$\boldsymbol{\alpha}^{(h)} \leftarrow \text{Softmax}\big((\boldsymbol{x} W_{q,h}^{\text{attn}})(\boldsymbol{x} W_{k,h}^{\text{attn}})^\top\big), \quad h \in [n_h^{sa}]$$
$$e_i^{(h)} \leftarrow \sum_j \alpha_{ij}^{(h)} x_j W_v^h, \quad i \in [n], h \in [n_h^{sa}]$$
$$e_i \leftarrow \text{concat}\big(e_i^{(1)}, \ldots, e_i^{(n_h^{sa})}\big) W_o^{sa}, \quad i \in [n]$$

```
Assign symbols:
```
$$\boldsymbol{s} = (s_1, \ldots, s_n) \leftarrow \text{SymbolRetriever}(\boldsymbol{x}; S_{\text{lib}})$$

```
Compute relational attention heads
```
$$\boldsymbol{\alpha}^{(h)} \leftarrow \text{Softmax}\big((\boldsymbol{x} W_{q,h}^{\text{attn}})(\boldsymbol{x} W_{k,h}^{\text{attn}})^\top\big), \quad h \in [n_h^{ra}]$$
$$\boldsymbol{r}_{ij} \leftarrow \big(\langle x_i W_{q,\ell}^{\text{rel}}, x_j W_{k,\ell}^{\text{rel}}\rangle\big)_{\ell \in [d_r]} \quad i, j \in [n]$$
$$a_i^{(h)} \leftarrow \sum_j \alpha_{ij}^{(h)}\big(\boldsymbol{r}_{ij} W_r^h + s_j W_s^h\big), \quad i \in [n], h \in [n_h^{ra}]$$
$$a_i \leftarrow \text{concat}\big(a_i^{(1)}, \ldots, a_i^{(n_h^{ra})}\big) W_o^{ra}, \quad i \in [n]$$

**Output:** $\big(\text{concat}(e_i, a_i)\big)_{i=1}^n$

---

## 3. Integrating Attention over Sensory and Relational Information

### 3.1. Dual Attention

One of the keys to the success of the Transformer architecture is the use of so-called *multi-head* attention. This involves computing multiple attention operations in parallel at each layer and concatenating the output, enabling the model to learn multiple useful criteria for routing information between objects. However, in standard Transformers, these attention heads focus solely on routing sensory information, lacking explicit support for routing *relational* information between objects.

We posit that both sensory and relational information are crucial for robust and flexible learning over sequences or collections of objects. To this end, we propose an extension of multi-head attention comprising two distinct types of attention heads: sensory attention (i.e., standard self-attention), and relational attention. This yields a powerful mechanism for dynamically routing both sensory and relational information in parallel. Our hypothesis is that by having access to both computational mechanism, the model can learn to select between them based on the current task or context, as well as compose them to create highly-expressive and flexible computational circuits.

Algorithm 1 describes the proposed module, referred to as *dual attention*. The number of sensory attention heads $n_h^{sa}$

**Algorithm 2:** Dual Attention Encoder Block

---

**Input :** $\boldsymbol{x} \in \mathbb{R}^{n \times d}$

$\boldsymbol{x} \leftarrow \text{Norm}(\boldsymbol{x} + \text{DualAttn}(\boldsymbol{x}))$
$\boldsymbol{x} \leftarrow \text{Norm}(\boldsymbol{x} + \text{MLP}(\boldsymbol{x}))$

**Output:** $\boldsymbol{x}$

---

**Algorithm 3:** Dual Attention Decoder Block

---

**Input :** $\boldsymbol{x}, \boldsymbol{y} \in \mathbb{R}^{n \times d}$

$\boldsymbol{x} \leftarrow \text{Norm}(\boldsymbol{x} + \text{DualAttn}(\boldsymbol{x}))$
$\boldsymbol{x} \leftarrow \text{Norm}(\boldsymbol{x} + \text{CrossAttn}(\boldsymbol{x}, \boldsymbol{y}))$
$\boldsymbol{x} \leftarrow \text{Norm}(\boldsymbol{x} + \text{MLP}(\boldsymbol{x}))$

**Output:** $\boldsymbol{x}$

---

and number of relational attention heads $n_h^{ra}$ are hyperparameters. The sensory attention heads attend to sensory information while the relational attention heads attend to relational information. The combined $n_h := n_h^{sa} + n_h^{ra}$ heads are then concatenated to produce the output. The result is a representation of contextual information with integrated sensory and relational components. Appendix B provides further discussion on the details of the architecture and its implementation.

**Attention Masks & Causality.** Any type of attention mask (e.g., causal mask for autoregressive language modeling) can be implemented in relational attention in the same way as for standard self-attention (i.e., mask is added to $\alpha_{ij}^h$ pre-softmax).

**Positional Encoding.** There exists different methods in the literature for encoding positional information in the Transformer architecture. For example, [25] propose adding positional embeddings to the input, [38] propose adding relative-positional embeddings at each attention operation, and [39] propose rotary positional embeddings (RoPE) which apply a position-dependent map to the queries and keys pre-softmax. These methods are compatible with dual attention and are configurable options in our public implementation.

**Computational complexity.** The computational complexity of relational attention scales similarly to standard self-attention with a $O(n^2)$ dependence on sequence length. Like standard attention, relational attention can be computed in parallel via efficient matrix multiplication operations.

**Symmetric relations.** A symmetry constraint can be injected into the relations $\boldsymbol{r}_{ij}$ by imposing that $W_q^{\text{rel}} = W_k^{\text{rel}}$, which is a useful inductive bias when the task-relevant relations are inherently symmetric.

### 3.2. The Dual Attention Transformer Architecture

The standard Transformer architecture is composed of repeated blocks of attention (information retrieval) followed by an MLP (local processing). Our proposed *Dual Attention Transformer* follows this same structure, but replaces multi-head self-attention with dual attention (Algorithm 1). At each layer, dual attention dynamically retrieves both sensory and relational information from the previous level of computation, which is then processed locally by an MLP. Al-

gorithms 2 and 3 in Appendix B define encoder and decoder blocks with dual attention. Composing these blocks yields the Dual Attention Transformer architecture.

The Dual Attention Transformer framework supports all architectural variants of the standard Transformer, making it applicable to a wide range of task paradigms. An encoder-decoder architecture with causal dual-head attention in the decoder can be applied to sequence-to-sequence tasks, as in the original Transformer paper [25]. An encoder-only architecture can be used for a BERT-style language embedding model [40] or a *ViT*-style vision model [41]. A decoder-only architecture with causal dual-head attention can be used for autoregressive language modeling.

## 4. Empirical evaluation

We empirically evaluate the Dual Attention Transformer (abbreviated, *DAT*) architecture on a range of tasks spanning different domains and modalities. Our goal is to assess the impact of integrating relational inductive biases into the Transformer architecture. We begin with a synthetic relational learning benchmark to evaluate *DAT*'s relational computational mechanisms in a more controlled setting. We then proceed to evaluate the proposed architecture on more complex real-world tasks, including mathematical problem-solving, image recognition, and language modeling. These experiments cover multiple task paradigms and architectural variants, including: discriminative (encoder-only architecture), sequence-to-sequence (encoder-decoder), autoregressive language modeling (decoder-only), and vision (*ViT*-style architecture) tasks. For each experiment, we compare a *DAT* model that incorporates both sensory and relational heads against a standard Transformer where all heads are ordinary sensory attention heads. The difference in performance highlights the impact of integrating both types of attention heads, enabling a richer representation of sensory and relational information. We summarize the experimental results below and defer certain experimental details to Appendix C.

### 4.1. Sample-Efficient Relational Reasoning: Relational Games

We begin our empirical evaluation with the "Relational Games" benchmark for visual relational reasoning proposed

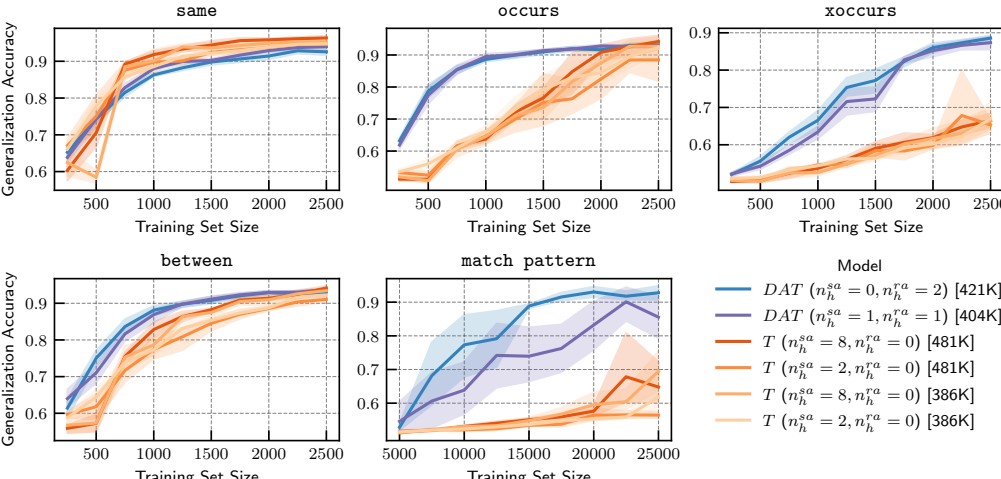

*Figure 2.* Learning curves on the relational games benchmark. Each subplot corresponds to a different task. Numbers in square brackets in legend labels indicate parameter counts. Solid lines indicate the mean over 5 trials with different random seeds and the shaded regions indicate bootstrap 95% confidence intervals. *DAT* is more data-efficient at relational learning compared to a Transformer.

by Shanahan et al. [20]. The dataset consists of a family of binary classification tasks, each testing a model's ability to identify a particular visual relationship among a series of objects (see Figure 5 for examples). The input is an RGB image depicting a grid of objects, and the target is a binary classification indicating whether the particular relationship holds for this input. This forms a controlled synthetic setting for evaluating *DAT*'s effectiveness in relational learning.

Our goal in this section is to explore how the relational computational mechanisms of *DAT* affect data-efficiency in relational learning—that is, how much data is necessary to learn a given task. We evaluate *learning curves* by varying the size of the training set, training each model until convergence, and evaluating on a hold-out validation set. We test two configurations of *DAT*: one with only relational attention heads, and one with a combination both sensory and relational heads. We include several Transformer baselines, varying the number of attention heads and the model dimension, controlling for parameter count. The results are depicted in Figure 2.

We find that *DAT* is significantly more sample-efficient, particularly at more difficult tasks. Both configurations of *DAT* are consistently more sample-efficient compared to the standard Transformer. The effect is particularly dramatic on the 'match pattern' task which is the most difficult and requires identifying a second-order relation (i.e., a relation between objects). We note that these tasks are purely relational in the sense that pairwise same/different relations between objects are a sufficient statistic for predicting the target. This suggests that relational attention is sufficient for solving the task. Indeed, the *DAT* variant with only relational heads performs slightly better than the variant with a combination of both sensory and relational heads. Notably,

however, the difference is only marginal, suggesting that the model is able to learn to select the computational mechanisms that are most relevant to the given task. We provide further discussion in Appendix C.1, including results comparing against previously-proposed relational architectures with stricter inductive biases.

## 4.2. Relational Inductive Biases for Symbolic Reasoning in Sequence-to-Sequence tasks: Mathematical Problem Solving

Next, we evaluate *DAT* on a set of mathematical problem-solving tasks based on the benchmark contributed by Saxton et al. [42]. Mathematical problem-solving is an interesting test for neural models because it requires more than statistical pattern recognition—it requires inferring laws, axioms, and symbol manipulation rules. The benchmark consists of a suite of mathematical problem-solving tasks, with each task's dataset consisting of a set of question-answer pairs. The tasks range across several topics including solving equations, adding polynomials, expanding polynomials, differentiating functions, predicting the next term in a sequence, etc. An example of a question in the "polynomials__expand" task is "Expand $(5x - 3)(2x + 1)$" with the target answer "$10x^2 - x - 3$". This is modeled as a sequence-to-sequence task with character-level encoding.

We compare *DAT* against Transformers using an encoder-decoder architecture. The encoder processes the question, and the decoder autoregressively generates the answer while cross-attending to the encoder. We explore how performance scales with model size by varying the number of layers. In the Transformer, all attention heads are standard self-attention with $n_h^{sa} = 8$, while in *DAT* we have a combination of both types of attention heads with $n_h^{sa} = n_h^{ra} = 4$.

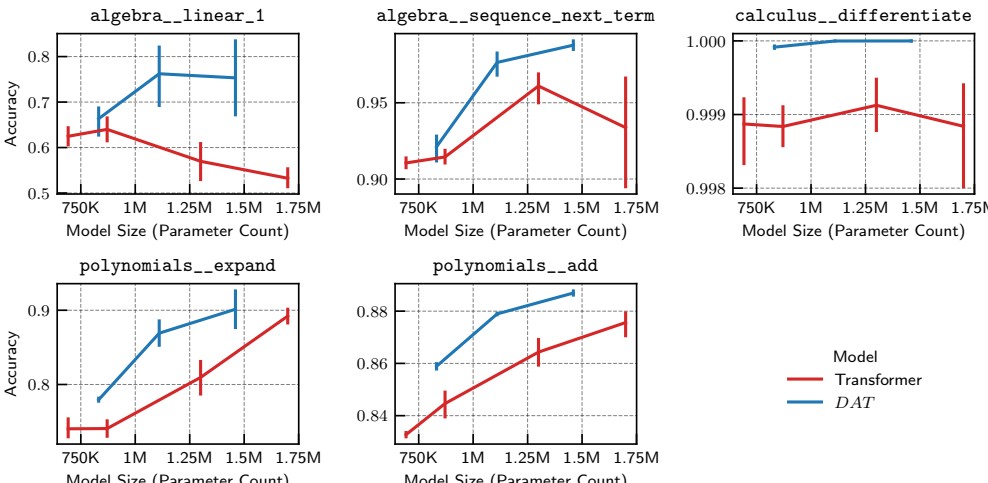

*Figure 3.* Average character-level accuracy on different mathematical problem-solving tasks measured at different model sizes. Error bars indicate bootstrap 95% confidence intervals over 5 trials. *DAT* outperforms a standard Transformer across model sizes, suggesting that relational computational mechanisms confer benefits on sequence-to-sequence tasks that involve symbolic computation.

Figure 3 depicts the character-level accuracy for *DAT* and Transformers across varying model sizes. We find that the *DAT* model outperforms the standard Transformer at all model scales and across all tested tasks. This suggests that the relational computational mechanisms of *DAT* are beneficial for the type of symbolic processing involved in solving mathematical problems.

### 4.3. Visual Processing with Relational Inductive Biases

As a general sequence model, the Transformer architecture can be applied to visual inputs by dividing an image into patches that are then flattened, linearly embedded into vectors, and passed in as a sequence. Through a series of attention and MLP operations, the visual input is processed for the downstream visual task. This architecture is referred to as a Vision Transformer (*ViT*) [41]. Although Transformers lack the explicit spatial inductive biases found in models like convolutional networks, recent work has demonstrated its effectiveness at scale [43], demonstrating the versatility of attention as a computational mechanism across several data modalities.

In this section, we explore how the relational computational mechanisms introduced in *DAT*—namely, relational attention—impact visual processing tasks. We hypothesize that visual processing benefits from attending to both sensory and relational information. That is, when processing a local region of a visual input (e.g., a patch, object, or object part), it is useful consider not only the sensory features of other regions but also the relationships between these regions. For example, this captures information about similar objects occurring in multiple places in the scene, or objects

which are similar across some attributes (e.g., texture) but different across others (e.g., color). In particular, the relations in relational attention can be interpreted as taking the source patch as a *filter* and comparing it against each patch in the image under different transformations.

We evaluate a *ViT*-style *DAT* architecture (*ViDAT*), and compare it against a standard *ViT* on the CIFAR image recognition benchmarks [44]. We train directly on CIFAR-10 and CIFAR-100, respectively, without pretraining on larger datasets. During training, we use random cropping, MixUp [45], and CutMix [46] as data augmentation techniques. We evaluate 8-layer models with $d_{\text{model}} = d_{\text{ff}} = 384$. The *ViT* model has $n_h^{sa} = 12$ standard self-attention heads, while the *DAT* model uses both sensory and relational heads, with an even split of $n_h^{sa} = n_h^{ra} = 6$. We use symmetric relations $\boldsymbol{r}_{ij}$ based on the intuition that visual processing involves symmetric attribute-similarity relations. In Appendix C.3, we present ablations, additional results, visualizations of learn relations, and further discussion.

Table 1 reports the classification accuracy of each model. *ViDAT* outperforms *ViT* on both datasets, suggesting that relational computational mechanisms enhance visual processing. These experiments show that relational inductive biases can be useful for image recognition. We hypothesize that relational processing is even more important in visual tasks requiring complex scene parsing, where reasoning about the relationships between constituent objects is essential. Recent work on scene understanding in large vision-language models supports this view [47–49]. We leave the exploration of such tasks for future work.

## 4.4. Relational Inductive Biases in Language Modeling

Language understanding involves processing and organizing relational information, such as syntactic structures, semantic roles, and contextual dependencies, to extract meaning from words and their connections within sentences. Transformers have been remarkably effective at language modeling, with neural scaling laws demonstrating that increasing model size and dataset size result in predictable improvements in performance across a range of language tasks [34, 35]. While the standard attention mechanism of Transformers is able to capture simple positional and syntactic relations in its attention scores, this is only used to control the flow of information between tokens rather than explicitly encoding relational information in the latent embeddings themselves. The relational attention mechanism of *DAT* enables explicitly learning relational contextual information that is directly encoded in each token's latent embedding.

In this section, we evaluate *DAT* on causal language modeling, exploring the impact of its relational computational mechanisms in the domain of language. We use a decoder-only architecture, where the model receives a sequence of tokens as input and is trained to causally predict the next token at each position. We train on 10 billion tokens of the FineWeb-Edu dataset [50], which is a curated dataset of high-quality educational text data from CommonCrawl. We train models at multiple parameter scales, up to 1.3 billion parameters, to study the scaling properties of *DAT* on language modeling with respect to both model size and data size. Details of training and architectural hyperparameters are given in Appendix C.4, together with further discussion of the results.

Figure 4 depicts the scaling properties of *DAT*'s language modeling performance with respect to model size and data size, compared to a standard Transformer. We observe that *DAT* demonstrates greater data and parameter efficiency, achieving improved performance across model and data scales. This suggests that *DAT*'s relational computational mechanisms confers benefits in language processing.

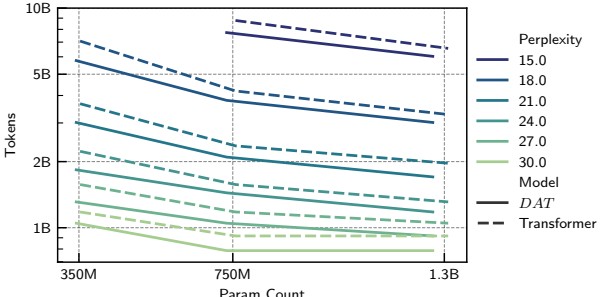

*Figure 4.* Performance scaling of *DAT* compared to a standard Transfomer on language modeling: *DAT* is more data-efficient and more parameter-efficient.

Beyond performance improvements, we also find evidence that relational attention encodes human-interpretable semantic relations. Figure 8 depicts a visualization of the relations $r_{ij}$ learned by a *DAT* language model. We observe that the relations learned by relational attention tend to encode semantic relations, rather than syntactic relations. That is, relational activations $r_{ij} \in \mathbb{R}^{d_r}$ are large between tokens with related *meanings*. We believe that further exploration of this phenomenon from a mechanistic interpretability perspective could offer an exciting avenue for future research. Such an exploration would be complementary to interpretability efforts seeking to understand the attention scores of standard Transformers, which have been found to attend e.g., based on position, syntax, and punctuation [51–53].

## 5. Conclusion

**Summary.** The standard attention mechanism of Transformers provides a versatile mechanism for retrieval of sensory information from a given context, but does not explicitly support retrieval of relational information. In this work, we presented an extension of the Transformer architecture that disentangles and integrates sensory and relational information through a variant of multi-head attention with two distinct types of attention heads: standard self-attention for sensory information and a novel *relational attention* mechanism for relational information. We empirically evaluate this architecture and find that it yields performance improvements across a range of tasks and modalities.

**Limitations & Future Work.** The proposed architecture introduces several hyperparameters and possible configurations. Although we carried out ablations on the major configuration choices (e.g., composition of head types, symmetry, symbol assignment mechanisms), an expanded empirical investigation would help develop an improved understanding of the behavior of this architecture under different configurations. We also note that our implementation of the Dual-Attention Transformer currently lacks the hardware-aware optimizations available for standard Transformers (e.g., Flash-Attention [54]), which makes it slower to train

| Dataset | Model | Params | Accuracy |
|---|---|---|---|
| CIFAR-10 | *ViT* | 7.1M | $86.4 \pm 0.1\%$ |
| | *ViDAT* | 6.0M | $\mathbf{89.7 \pm 0.1\%}$ |
| CIFAR-100 | *ViT* | 7.2M | $68.8 \pm 0.2\%$ |
| | *ViDAT* | 6.1M | $\mathbf{70.5 \pm 0.1\%}$ |

*Table 1.* Classification accuracy on image recognition with the CIFAR-10 and CIFAR-100 datasets. Each training configuration is repeated 10 times with different random seeds; we report the mean accuracy $\pm$ the standard error of mean. *DAT* outperforms a standard Vision Transformer, suggesting that relational computational mechanisms are useful for visual processing tasks.

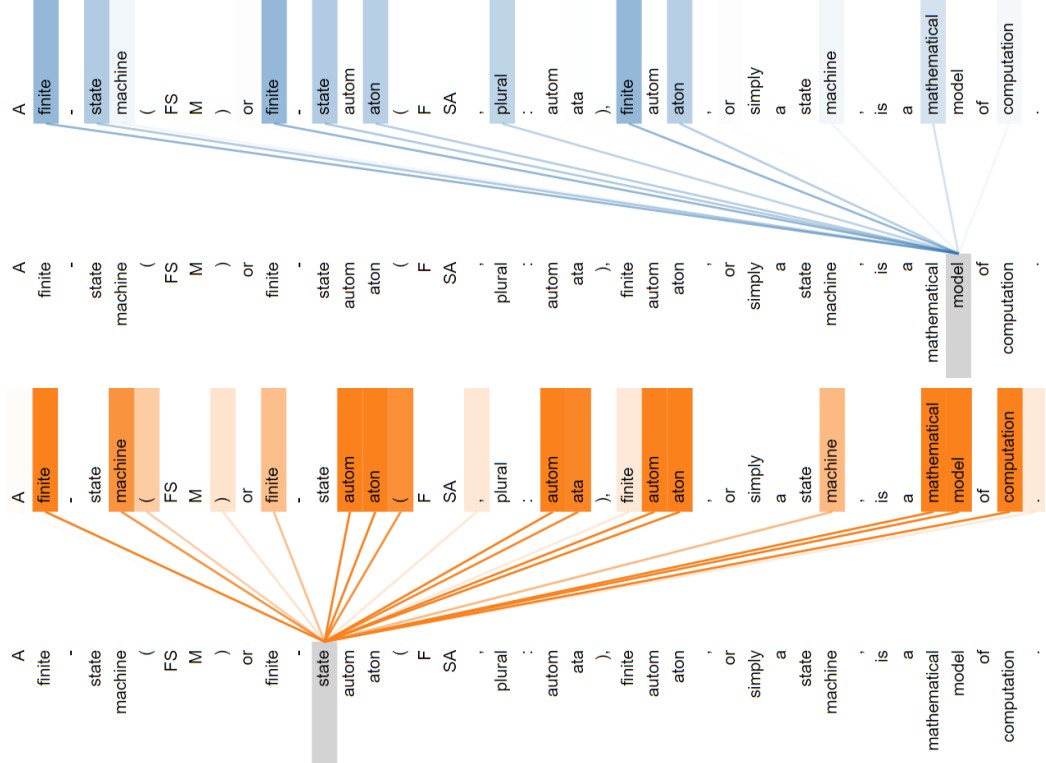

Figure 5. Relational attention in *DAT* language models encodes human-interpretable semantic relations. A visualization of the relations $r_{ij}$ learned by a 24-layer 343M-parameter *DAT* language model. **Top.** Visualization of one relation dimension in the first layer, focusing on the token 'model', which has high activation with the tokens 'state', 'machine', and 'mathematical'. **Bottom.** Visualization of one relation dimension in the twelfth layer, focusing on the token 'state', which has high activation with the tokens 'mathematical', 'model', and 'computation'.

overall (though we expect similar optimizations to be possible). An important direction for future work is the mechanistic interpretability [53, 55, 56] of *DAT* models, focusing on identifying specific circuits that perform key computations, to better understand the performance improvements observed in complex domains like language modeling.

## Code and Reproducibility

Our implementation of the Dual Attention Transformer architecture is open-sourced at https://github.com/Awni00/dual-attention and published as a Python package. Pre-trained model weights, including the 1.3B-parameter *DAT* language model, are made publicly available and can be loaded directly using the package. Additionally, we provide code for running the experiments described in this paper, along with instructions for reproducing our results and access to the experimental logs.

## Acknowledgment

This research was supported by the funds provided by the National Science Foundation and by DoD OUSD (R&E) under Cooperative Agreement PHY-2229929 (The NSF AI Institute for Artificial and Natural Intelligence).

## Impact Statement

This work develops a variant of the Transformer architecture, called the Dual Attention Transformer, that integrates explicit relational computational mechanisms to enhance data-efficiency and generalization in learning relational processing. This advancement has the potential to improve performance in applications requiring structured reasoning. Like other developments in Transformer architectures, this research shares the potential societal consequences associated with the broader field.

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

## A. Function Class of Relational Attention: a universal approximation result

To gain a better understanding of the types of functions that can be computed by relational attention, we presented a simple approximation result (Theorem 1) in Section 2.4. Here, we will provide a formal statement of the result and prove it.

Recall that relational attention is a mapping on $\mathbb{R}^d \times \mathbb{R}^{n \times d} \to \mathbb{R}^{d_{\text{out}}}$, where $d$ is the dimension of the input objects and $d_{\text{out}}$ is the output dimension. For convenience, we denote the "query space" by $\mathcal{X}$ and the "key space" by $\mathcal{Y}$, though both are $\mathbb{R}^d$ in this setting. Relational attention takes as input a query $x \in \mathcal{X}$ and a collection of objects $\boldsymbol{y} = (y_1, \ldots, y_n) \in \mathcal{Y}^n$ and computes the following

$$\mathrm{RA}(x, \boldsymbol{y}) = \sum_{i=1}^{n} \alpha_i(x; \boldsymbol{y})\big(r(x, y_i)\, W_r + s_i\, W_s\big), \tag{4}$$

$$\alpha(x; \boldsymbol{y}) = \mathrm{Softmax}\Big(\big[\, \langle \phi_q^{\text{attn}}(x), \phi_k^{\text{attn}}(y_i) \rangle \,\big]_{i=1}^{n}\Big) \in \Delta^n, \tag{5}$$

$$r(x, y_i) = \big(\langle \phi_{q,\ell}^{\text{rel}}(x), \phi_{k,\ell}^{\text{rel}}(y_i) \rangle\big)_{\ell \in [d_r]} \in \mathbb{R}^{d_r}, \tag{6}$$

$$(s_1, \ldots, s_n) = \mathrm{SymbolRetriever}\,(\boldsymbol{y}; S_{\text{lib}}) \in \mathbb{R}^{n \times d_{\text{out}}}, \tag{7}$$

where $\phi_q^{\text{attn}}, \phi_k^{\text{attn}}, \phi_{q,\ell}^{\text{rel}}, \phi_{k,\ell}^{\text{rel}} : \mathbb{R}^d \to \mathbb{R}^{d_k}$ are the feature maps defining the attention mechanism and the relation, respectively. For this section, these are multi-layer perceptrons. Note that in Algorithm 1 these are linear maps, but they are preceded by multi-layer perceptron in Algorithms 2 and 3, which makes the overall function class the same. Moreover, for this analysis we will take $W_r = I, d_{\text{out}} = d_r$ and $W_s = 0$. We will later discuss how the role of symbols fits within the message of the result.

The following result states that relational attention can approximate any function of the form: 1) select an object in $(y_1, \ldots, y_n)$ by an arbitrary query-dependent selection criterion, and 2) compute an arbitrary relation $r : \mathcal{X} \times \mathcal{Y} \to \mathbb{R}^{d_r}$ with the selected object. This is formalized below.

To formalize (1), we adopt an abstract and very general formulation of a "selection criterion" in terms of a family of preference preorders, $\{\preccurlyeq_x\}_x$: for each possible query $x$, the preorder $\preccurlyeq_x$ defines a preference over objects in $\mathcal{Y}$ to be selected. Intuitively, "$y_1 \preccurlyeq_x y_2$" means that $y_2$ is more relevant to the query $x$ than $y_1$.

More precisely, for each query $x \in \mathcal{X}$, $\preccurlyeq_x$ is a complete (for each $y_1, y_2 \in \mathcal{Y}$, either $y_1 \preccurlyeq y_2$ or $y_2 \preccurlyeq_x y_1$), reflexive ($y \preccurlyeq_x y$ for all $y \in \mathcal{Y}$), and transitive ($y_1 \preccurlyeq_x y_2$ and $y_2 \preccurlyeq_x y_3$ implies $y_1 \preccurlyeq_x y_3$) relation. For each $x \in \mathcal{X}$, $\preccurlyeq_x$ induces a preordered space $(\mathcal{Y}, \preccurlyeq_x)$. This implicitly defines two additional relations: $\prec_x$ and $\sim_x$. We will write $y_1 \prec_x y_2$ if "$y_1 \preccurlyeq_x y_2$ and not $y_2 \preccurlyeq_x y_1$", and $y_1 \sim y_2$ if "$y_1 \preccurlyeq_x y_2$ and $y_2 \preccurlyeq_x y_1$".

For a collection of objects $\boldsymbol{y} = (y_1, \ldots, y_n) \in \mathcal{Y}^n$ and a query $x \in \mathcal{X}$, the preorder $\preccurlyeq_x$ defines a selection function

$$\mathrm{Select}(x, (y_1, \ldots, y_n)) \coloneqq \max\left((y_1, \ldots, y_n), \texttt{key} = \preccurlyeq_x\right). \tag{8}$$

That is, $\mathrm{Select}(x, \boldsymbol{y})$ returns the most relevant element with respect to the query $x$. In particular, it returns $y_i$ when $y_i \succ_x y_j, \ \forall j \neq i$ (and may return an arbitrary element if no unique maximal element exists in $(y_1, \ldots, y_n)$).

We will assume some regularity conditions on the family of preorders $\{\preccurlyeq_x\}_x$ which essentially stipulate that: 1) nearby elements in $\mathcal{Y}$ have a similar preference with respect to each $x$, and 2) nearby queries in $\mathcal{X}$ induce similar preference preorders.

**Assumption 1** (Selection criterion is query-continuous and key-continuous). *The family of preorder relations $\{\preccurlyeq_x\}_{x \in \mathcal{X}}$ satisfies the following:*

1. ***Key-continuity.*** *For each $x \in \mathcal{X}$, $\preccurlyeq_x$ is continuous. That is, for any sequence $(y_i)_i$ such that $y_i \preccurlyeq_x z$ and $y_i \to y_\infty$, we have $y_\infty \preccurlyeq_x z$. Equivalently, for any $y \in \mathcal{Y}$, $\{z \in \mathcal{Y} : z \preccurlyeq_x y\}$ and $\{z \in \mathcal{Y} : y \preccurlyeq_x z\}$ are closed sets in $\mathcal{Y}$.*

2. ***Query-continuity.*** *Under key-continuity, Debreu et al. [57] shows that for each $x \in \mathcal{X}$, there exists a continuous in utility function $u_x : \mathcal{Y} \to \mathbb{R}$ for $\preccurlyeq_x$ such that $y_1 \preccurlyeq_x y_2 \iff u_x(y_1) \leq u_x(y_2)$. For query-continuity, we make the further assumption that there exists a family of utility functions $\{u_x : \mathcal{Y} \to \mathbb{R}\}_{x \in \mathcal{X}}$ such that $u(x, y) \coloneqq u_x(y)$ is also continuous in its first argument.*

For technical reasons, for Equation (8) to make sense, we must assume that there exists a unique element to be selected. We formulate this in terms of an assumption on the data distribution of the space $\mathcal{X} \times \mathcal{Y}^n$. This is a technical assumption, and different forms of such an assumption would be possible (e.g., instead condition on this event).

**Assumption 2** (Selection is unique almost always). *Let $(x, \boldsymbol{y}) \sim \mathbb{P}_{x,\boldsymbol{y}}$. For each $\varepsilon > 0$, there exists $\eta_\varepsilon > 0$ such that* $\min_{j \neq i} |u_x(y_i) - u_x(y_j)| > \eta_\varepsilon$ *with probability at least $1 - \varepsilon$.*

**Theorem** (Function class of relational attention). *Let $\mathcal{X}, \mathcal{Y}$ be compact Euclidean spaces. Let $\{\preccurlyeq_x\}_{x \in \mathcal{X}}$ be an arbitrary family of relevance preorders on $\mathcal{Y}$ which are query-continuous and key-continuous (Assumption 1). Let* $\mathrm{Select}(x, (y_1, \ldots, y_n)) = \max((y_1, \ldots, y_n), \mathtt{key} =\preccurlyeq_x)$ *be the selection function associated with $\{\preccurlyeq_x\}_x$. Let $R : \mathcal{X} \times \mathcal{Y} \to \mathbb{R}^{d_r}$ be an arbitrary continuous relation function. Suppose $x, \boldsymbol{y} \sim \mathbb{P}_{x,\boldsymbol{y}}$ and that Assumption 2 holds (i.e., the data distribution is such that there exists a unique most-relevant element w.h.p). For any $\varepsilon > 0$, there exists multi-layer perceptrons* $\phi_q^{\mathrm{attn}}, \phi_k^{\mathrm{attn}}, \phi_q^{\mathrm{rel}}, \phi_k^{\mathrm{rel}}$ *and a choice of symbols such that,*

$$\|\mathrm{RA}(x, (y_1, \ldots, y_n)) - R(x, \mathrm{Select}(x, (y_1, \ldots, y_n)))\|_\infty < \varepsilon$$

*Proof.* Condition on the event $\mathcal{E} := \{(x, \boldsymbol{y}) \in \mathcal{X} \times \mathcal{Y}^n : \min_{j \neq i} |u_x(y_i) - u_x(y_j)| > \eta_\varepsilon\}$. Let $i^* = \arg\max((y_1, \ldots, y_n), \mathtt{key} =\preccurlyeq_x) = \arg\max(u_x(y_1), \ldots, u_x(y_n))$. By [58, Theorem 5.1], for any $\varepsilon_1 > 0$, there exists MLPs $\phi_q^{\mathrm{attn}}, \phi_k^{\mathrm{attn}}$ such that $\alpha_{i^*}(x, \boldsymbol{y}) > 1 - \varepsilon_1$ for any $(x, \boldsymbol{y}) \in \mathcal{E}$. That is, the attention score is nearly 1 for the $\preccurlyeq_x$-selected element *uniformly* over inputs in $\mathcal{E}$.

Similarly, by [58, Theorem 3.1], for any $\varepsilon_2 > 0$, there exists MLPs $(\phi_{q,\ell}^{\mathrm{rel}}, \phi_{k,\ell}^{\mathrm{rel}})_{\ell \in [d_r]}$ such that $r(x, y) := (\langle \phi_{q,\ell}^{\mathrm{rel}}(x), \phi_{k,\ell}^{\mathrm{rel}}(y) \rangle)_{\ell \in [d_r]}$ approximates the target relation $R$ uniformly within an error of $\varepsilon_2$,

$$\|R(x, y) - r(x, y)\|_\infty < \varepsilon_2, \quad \text{Lebesgue almost every } (x, y) \in \mathcal{X} \times \mathcal{Y}.$$

Thus, we have

$$\|\mathrm{RA}(x, (y_1, \ldots, y_n)) - R(x, \mathrm{Select}(x, (y_1, \ldots, y_n)))\|_\infty$$
$$= \left\| \sum_{i=1}^n \alpha_i(x; \boldsymbol{y}) \, r(x, y_i) - R(x, y_{i^*}) \right\|_\infty$$
$$\leq \sum_{i=1}^n \|\alpha_i(x; \boldsymbol{y}) \, r(x, y_i) - R(x, y_{i^*})\|_\infty$$
$$\leq \alpha_{i^*}(x, \boldsymbol{y}) \|r(x, y_{i^*}) - R(x, y_{i^*})\|_\infty + \sum_{j \neq i^*} \alpha_i(x; \boldsymbol{y}) \|r(x, y_i) - R(x, y_{i^*})\|_\infty$$
$$\leq (1 - \varepsilon_1)\varepsilon_2 + \varepsilon_1 \max_{x,y,y^*} \|r(x, y) - R(x, y^*)\|_\infty.$$

Note that $\max_{x,y,y^*} \|r(x, y) - R(x, y^*)\|_\infty$ is finite since $\mathcal{X}, \mathcal{Y}$ are compact and $r, R$ are continuous. Letting $\varepsilon_1, \varepsilon_2$ be small enough completes the proof. $\qquad\square$

To summarize the analysis in this section, we showed that relational attention can approximate any computation composed of first selecting an object from a collection then computing a relation with that object. We can approximate any well-behaved selection criterion by formulating it in terms of an abstract preference preorder, and approximating the corresponding utility function (given by a Debreu representation theorem) by inner products of query and key feature maps. We can then approximate the target relation function similarly by inner products of a different set of query and key feature maps.

In the analysis above, we set aside the role of the symbols. Note that the function class this approximation result proves involves retrieving a relation from a selected object, but does not explicitly encode the identity of the selected object. Informally, the receiver knows that it has a particular relation with one of the objects in its context, and knows that this relation is with an object that was selected according to a particular selection criterion, but does not know the identity of the object beyond that. This is the purpose of adding symbols to relational attention—the retrieved relation is tagged with a symbol identifying the source.

# B. Architecture & Implementation Details

In this section, we briefly discuss some details of implementation that may be of interest to some readers. Our code is publicly available through the project git repository and includes detailed instructions for reproducing our experimental results. We also provide links to experimental logs. Our code uses the PyTorch framework.

## B.1. Relational Attention and Dual-Head Attention

The relational attention operation is defined as part of dual-head attention in Algorithm 1. We briefly mention some details of the implementation.

***Learnable parameters.*** Let $n_h := n_h^{sa} + n_h^{ra}$ be the total number of sensory and relational heads. The learnable parameters are

- **Sensory attention heads.** For each head $h \in [n_h^{sa}]$:
  - Attention query/key projections: $W_{q,h}^{\text{attn}}, W_{k,h}^{\text{attn}} \in \mathbb{R}^{d_{\text{model}} \times d_{\text{key}}}$,
  - Value projections: $W_v^h \in \mathbb{R}^{d_{\text{model}} \times d_h}$,
  - Output projection: $W_o^{sa} \in \mathbb{R}^{d_{\text{model}} \times d_{\text{model}}}$.

- **Relational attention heads.** For each head $h \in [n_h^{ra}]$ and each relation $\ell \in [d_r]$:
  - Attention query/key projections: $W_{q,h}^{\text{attn}}, W_{k,h}^{\text{attn}} \in \mathbb{R}^{d_{\text{model}} \times d_{\text{key}}}$,
  - Relation query/key projections: $W_{q,\ell}^{\text{rel}}, W_{k,\ell}^{\text{rel}} \in \mathbb{R}^{d_{\text{model}} \times d_{\text{proj}}}$,
  - Symbol projection: $W_s^h \in \mathbb{R}^{d_{\text{model}} \times d_h}$,
  - Relation projection: $W_r^h \in \mathbb{R}^{d_r \times d_h}$,
  - Output projection: $W_o^{ra} \in \mathbb{R}^{d_{\text{model}} \times d_{\text{model}}}$.

We let $d_{\text{key}}, d_h = d_{\text{model}}/n_h$ to maintain the same dimension for the input and output objects. Similarly, we let $d_{\text{proj}} = d_h \cdot n_h^{ra}/d_r$ so that the number of parameters is fixed as $d_r$ varies. That is, we scale $d_{\text{proj}}$ down as $d_r$ increases; $d_{\text{proj}}$ has the interpretation of being the dimensionality of the subspace on which we are computing comparisons. So, having a larger number of relations corresponds to a more fine-grained comparison between the two objects.

To model symmetric relations, we let $W_{q,\ell}^{\text{rel}} = W_{k,\ell}^{\text{rel}}$. Recall that this has the interpretation of computing a comparison between the same attributes in the pair of objects.

Note that the same $d_r$-dimensional relation is used for all $n_h^{ra}$ attention heads, with a different learned linear map $W_r^h$ for each head extracting the relevant aspects of the relation for that attention head and controlling the placement in the residual stream. This allows for useful computations to be shared across all heads. Note also that the head dimension $d_h = d_{\text{model}}/n_h$ is defined in terms of the total number of attention heads and is the same for both sensory attention and relational attention. The output of each head is a $d_h$-dimensional vector. This means that after concatenating all the heads, the proportion in the final $d_{\text{model}}$-dimensional output that corresponds to each attention head type is proportional to the number of heads of that type. For example, if $n_h^{sa} = 6, n_h^{ra} = 2$, then 75% of the $d_{\text{model}}$-dimensional output is composed of the output of sensory attention heads and 25% is composed of the output of relational attention heads. This enables tuning the relative importance of each head type for the task.

***Code.*** We briefly discuss the code implementing relational attention. We use `einsum` operations heavily in our implementation due to the flexibility they offer for implementing general tensor contractions. From Algorithm 1, recall that relational attention takes the form:

$$a_i^{(h)} \leftarrow \sum_j \alpha_{ij}^{(h)} \big( \boldsymbol{r}_{ij} W_r^h + s_j W_s^h \big), \tag{9}$$

where $\alpha_{ij}^{(h)}$ are the softmax attention scores for head $h \in [n_h^{ra}]$, $\boldsymbol{r}_{ij} \in \mathbb{R}^{d_r}$ are relation vectors, $s_j \in \mathbb{R}^{d_{\text{model}}}$ is the symbol associated with the $j$-th input, and $W_r^h, W_s^h$ map $\boldsymbol{r}_{ij}$ and $s_j$, respectively, to $d_h$-dimensional vectors. We assume those are already computed and focus on a particular portion of the computation of relational attention. We break up the computation as follows:

$$\sum_j \alpha_{ij}^{(h)} \big( \boldsymbol{r}_{ij} W_r^h + s_j W_s^h \big) = \sum_j \big( \alpha_{ij}^{(h)} s_j W_s^h \big) + \big( \sum_j \alpha_{ij}^{(h)} \boldsymbol{r}_{ij} \big) W_r^h. \tag{10}$$

Note that we factor out the $W_r^h$ linear map and apply it after computing $\sum_j \alpha_{ij}^{(h)} \boldsymbol{r}_{ij}$. This is intentional, as will be explained below.

This can be computed in PyTorch via `einsum` operations as follows.

```python
# sv: (b, n, n_h, d_h)
# attn_scores: (b, n_h, n, n)
# relations: (b, n, n, d_r)
# self.wr: (n_h, d_h, d_r)

attended_symbols = torch.einsum('bhij,bjhd->bihd', attn_scores, sv)
# shape: (b, n, n_h, d_h)

attended_relations = torch.einsum('bhij,bijr->bihr', attn_scores, relations)
# shape: (b, n, n_h, d_r)

attended_relations = torch.einsum('bihr,hdr->bihd', attended_relations, self.wr)
# shape: (b, n, n_h, d_h)

output = attended_symbols + attended_relations
# shape: (b, n, n_h, d_h)
```

Here, we assume `sv`, `attn_scores`, and `relations` are already computed, and focus on a particular part of the computation. `sv[:,:,h,:]` $= \boldsymbol{s}\, W_s^h$, corresponds to the symbols of each object in the context, `attn_scores[:,h,:,:]` $= \boldsymbol{\alpha}^h$ are the softmax attention scores, and `relations[:, i,j,:]` $= \boldsymbol{r}_{ij}$ are the relations, which can all be computed with simple matrix multiplication operations, very similar to the standard implementations of multi-head attention.

The first line corresponds to computing $\sum_j \alpha_{ij}^h s_j W_s^h$. The second line corresponds to computing $\sum_j \alpha_{ij}^h \boldsymbol{r}_{ij}$. The third line corresponds to applying the linear map $W_r^h$ to the retrieved relations at each head. The reason we apply the map $W_r^h$ after attending to the relations is for memory efficiency reasons. If we were to apply $W_r^h$ first, we would need to manifest a tensor of dimension $b \times n \times n \times n_h^{ra} \times d_h$, which is of order $\mathcal{O}(b \cdot n^2 \cdot d_{\text{model}})$. Instead, by factoring out $W_r^h$ and applying it after computing attention, we only need to manifest a tensor of dimension $b \times n \times n \times d_r$, which is much smaller since $d_r \ll d_{\text{model}}$. This tensor is contracted to a dimension $b \times n \times d_r$ first, *then* mapped up to $b \times n \times n_h^{ra} \times d_h$. This makes the memory footprint of relational attention of the same order as standard (sensory) attention when $d_r \asymp n_h$.

When using position-relative symbols, the implementation is adjusted since we need to compute

$$\sum_j \alpha_{ij}^{(h)} \left( \boldsymbol{r}_{ij} W_r^h + s_{j-i} W_s^h \right) \tag{11}$$

instead, where the symbol $s_{j-i}$ sent now depends on both the source $j$ and the target $i$. Thus, we now compute a symbols tensor which is indexed by both the source $j$ and target $i$: `sv[i,j,h,:]` $= s_{j-i} W_s^h$. Then, the implementation is adjusted by replacing the first line in the code above with

```python
attended_symbols = torch.einsum('bhij,ijhd->bihd', attn_scores, sv)
```

The full implementation is made available through the project's github repository.

***Composing relational attention to learn hierarchical relations.*** We remark that composing relational attention modules can be interpreted as representing hierarchical or higher-order relations. That is, relations between relations. An example of this is the relation tested in the `match pattern` task in the relational games benchmark. After one iteration of relational attention, an object's representation is updated with the relations it has with its context. A second iteration of relational attention now computes a representation of the relation between an object's relations and the relations of the objects in its context.

## B.2. Encoder and Decoder Blocks

We briefly mention a few configurations in our implementation that appear in our experiments. We aimed to make our implementation configurable to allow for various tweaks and optimizations that have been found in the literature for training Transformer models.

**Symbol assignment.** A shared symbol assignment module is used for all layers in the model. We explore three types of symbol assignment mechanisms: positional symbols, position-relative symbols, and symbolic attention. Different symbol assignment mechanisms are more well-suited to different tasks. We discuss ablation experiments we carried out on the effect of the symbol assignment mechanism in Appendix C.

**MLP block.** The MLP block uses a 2-layer feedforward network with a configurable activation function. The intermediate layer size is $d_{\text{ff}} = 4 \cdot d_{\text{model}}$ by default. We also use the SwiGLU "activation function" [59] in some of our experiments. SwiGLU is not merely an activation function, but is rather a neural network layer defined as the component-wise product of two linear transformations of the input. It is a type of gated linear unit [60] with the sigmoid activation replaced with a Swish activation [61], $\text{SwiGLU}(x) = \text{Swish}(xW + b) \otimes (xV + c)$. This is used in the Llama series of models and was found to be a useful modification [62].

**Normalization.** Either LayerNorm [63] or RMSNorm [64] can be used. Normalization can be performed post-attention, like in the original Transformer paper [25], or pre-attention as in [65].

**Positional encoding.** Our experiments use either learned positional embeddings or RoPE [39].

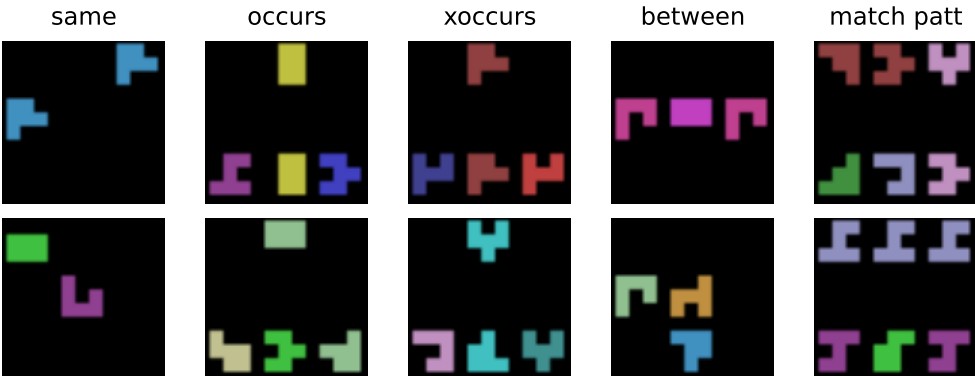

*Figure 6.* Examples of different tasks in the Relational Games benchmark. Each column corresponds to a different task in the benchmark. The top row is an example of a positive instance and the bottom row is an example of a negative instance.

## C. Experimental Details & Further Discussion

### C.1. Relational Games (Section 4.1)

EXPERIMENTAL DETAILS

**Dataset details.** The Relational Games benchmark datasets consists of $36 \times 36 \times 3$ RGB images depicting a $3 \times 3$ grid of objects which satisfy a particular visual relationship. The task is to identify whether a given relationship holds or not. The set of objects consists of simple geometric shapes. Examples of each task are presented in Figure 5. For example, in the `occurs` task, one object is present in the top row and three in the bottom row, and the task is to determine whether the object in the top row occurs (i.e., is among) the objects in the bottom row. The most difficult task in the benchmark is the `match pattern` task, where the grid contains a triplet of objects in the top row and another triplet of objects in the bottom row. Each triplet satisfies some relationship (e.g., ABC, ABA, ABB, or AAB), and the task is to determine whether the relation in the first triplet is the same as the relation in the second triplet. The difficulty in solving this task is that it requires parsing a second-order relation (a relation between relations). We remark that composing relational attention modules naturally captures this kind of hierarchical relations: the first relational attention operation produces objects representing relational information and the second would compute relations between those relations (i.e., second-order relations).

**Model architectures.** We use a Vision-Transformer-type architecture where the input image is split up into patches, flattened, and passed through the sequence model with added learned positional embeddings. We use average pooling at the end and pass through an MLP to produce the final prediction. We use a patch size of $12 \times 12$ which separates objects according to the grid structure. We note that in more general visual relational reasoning tasks where there isn't this type of grid structure, it would be appropriate to combine our approach with an object-discovery module such as Slot Attention [66].

We use 2-layer models. The *DAT* models use $d_{\text{model}} = 128$, $d_{\text{ff}} = 256$. One set of Transformer baselines uses the same, while another is larger with $d_{\text{model}} = 144$, $d_{\text{ff}} = 288$. All models use SwiGLU "activation", dropout rate = 0.1, and pre-LayerNormalization. For the *DAT* models, we use positional symbols as the symbol assignment mechanism. The composition of sensory and relational attention heads are depicted in the figure. In Figure 2, we use symmetric relations (i.e., imposing that $W_q^{\text{rel}} = W_k^{\text{rel}}$). Below, we also explore the effect of this inductive bias, evaluating variants without the symmetry constraint.

**Training details.** For each task and model, we evaluated learning curves by varying the training set size and training the model until convergence, then evaluating on a hold-out test set. For four out of five of the tasks, we evaluate learning curves within the range of 250 to $2,500$ samples, in increments of 250. For the more difficult `match pattern`, the range is from $5,000$ to $25,000$ in increments of $5,000$. The ranges were chosen based on the difficulty of the different tasks in order to identify the right "resolution". When evaluating learning curves, each training set is sampled randomly from the full dataset. For each task, model, and training set size, we repeat the experiment 5 times with different random seeds to compute approximate confidence intervals (accounting for randomness in sampling the dataset and random initialization). We use an Adam optimizer with a learning rate of 0.001, $\beta_1 = 0.9$, $\beta_2 = 0.99$, and a batch size of 512. We train for 50 epochs.

**Comparison to previous relational architectures.** Previous research has explored relational learning in synthetic settings, proposing various architectures with relational inductive biases. Here, we compare *DAT* to three such architectures: PrediNet [20], CoRelNet [21], and Abstractor [22]. Unlike *DAT*, these architectures use subtractive rather than additive relational inductive biases, imposing constraints on the types of learnable representations to improve relational learning efficiency. As a result, they are not general-purpose architectures and cannot be applied to broader domains such as language modeling. Nonetheless, it is useful to compare *DAT* against those architectures to explore the trade-offs of strong inductive biases and evaluate *DAT* in comparison to alternative approaches to relational learning. Figure 6 shows learning curves comparing *DAT* against those baselines. *DAT* performs competitively with previous relational architectures, generally outperforming PrediNet and Abstractor, while performing marginally worse than CoRelNet. It is relevant to note that CoRelNet incorporates strong task-specific inductive biases, and was partially designed with this benchmark in mind.

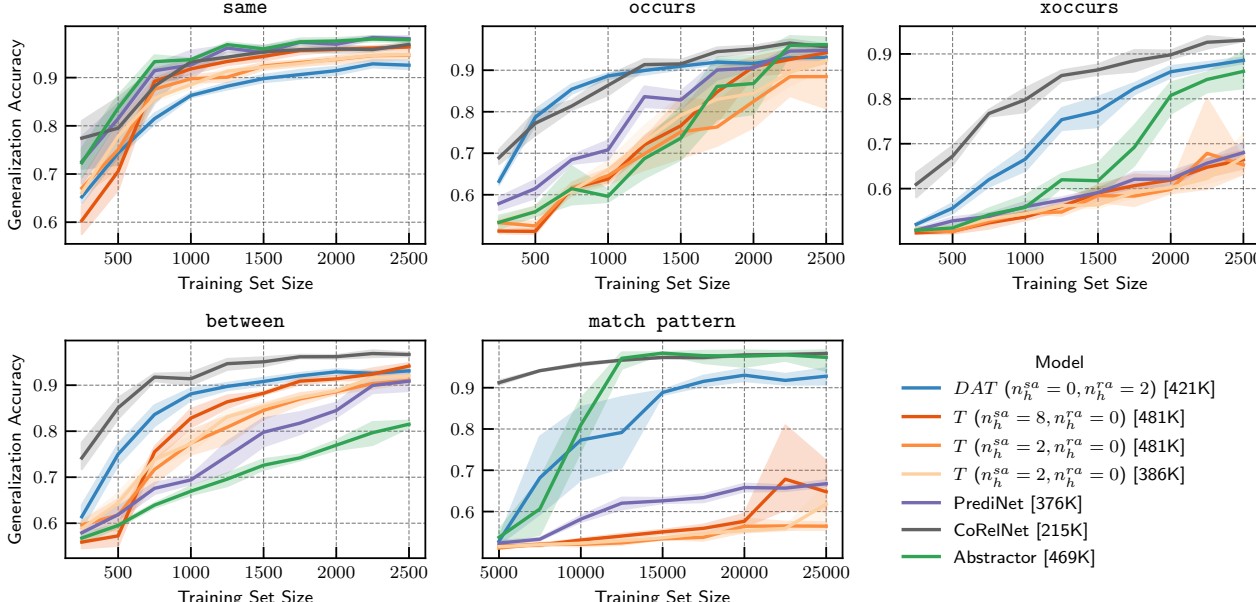

*Figure 7.* Learning curves on the Relational Games benchmark, comparing *DAT* against previously-proposed relational architectures. *DAT* performs competitively with previous relational architectures.

**Ablation over symmetry.** We performed an ablation over the *symmetry* inductive bias in the relations computed in relational attention. Our implementation exposes an argument which controls whether the relation $r(x, y) = (\langle W_{q,\ell}^{\mathrm{rel}}, W_{k,\ell}^{\mathrm{rel}} \rangle)_{\ell \in [d_r]} \in \mathbb{R}^{d_r}$ modeled in relational attention is constrained to be symmetric by setting $W_{q,\ell}^{\mathrm{rel}} = W_{k,\ell}^{\mathrm{rel}}$. Indeed, we find symmetry to be a useful inductive bias in this task. Figure 7 depicts learning curves for the two configurations of *DAT* comparing symmetric RA against asymmetric RA. We find that symmetry results in faster learning curves for both configurations.

## C.2. Mathematical Problem-Solving (Section 4.2)

EXPERIMENTAL DETAILS

**Dataset details.** Saxton et al. [42] propose a benchmark to assess neural models' ability to perform mathematical reasoning. The dataset consists of a suite of tasks in free-form textual input/output format. The tasks cover several topics in mathematics, including arithmetic, algebra, and calculus. For each task, the authors programmatically generate $2 \times 10^6$ training examples and $10^4$ validation examples. Questions have a maximum length of 160 characters and answers have a maximum length of 30 characters.

**Model architectures.** We use an encoder-decoder architecture for this experiment, treating it as a sequence-to-sequence task. We use character-level encoding with a common alphabet of size 85 containing small and upper case letters, digits 0-9, and symbols (e.g., `*`, `/`, `+`, `-`). We vary the number of layers to explore how performance scales with model size

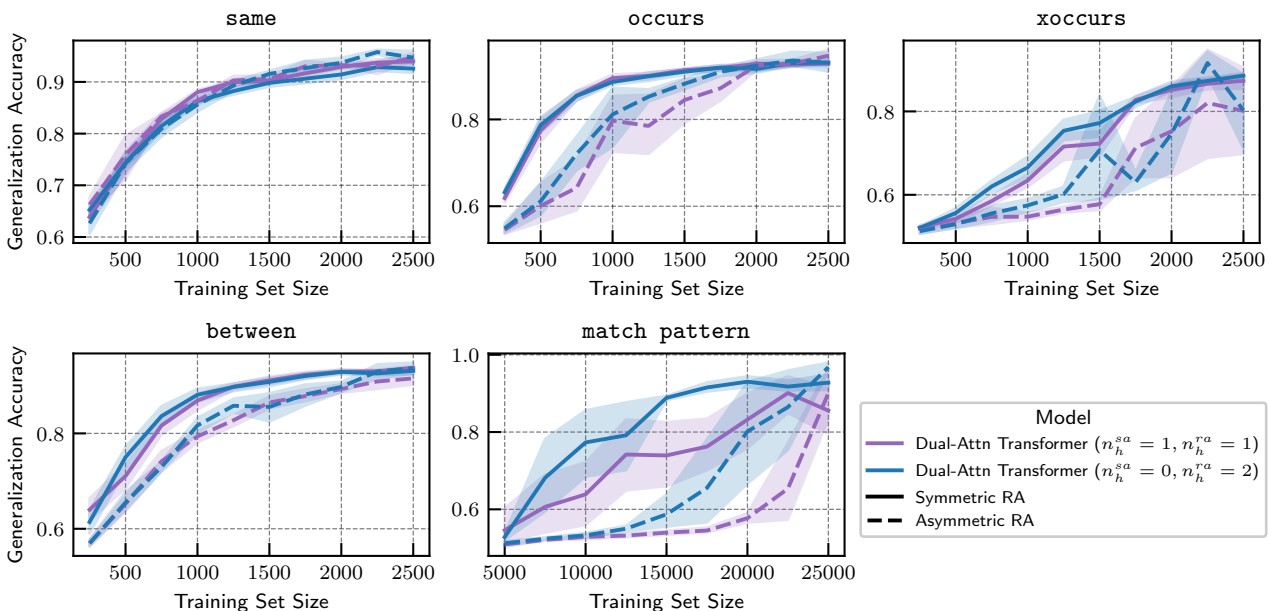

*Figure 8.* An ablation of the effect of symmetry in relational attention in the relational games experiments.

in *DAT* compared to standard Transformers. Each encode/decoder block uses ReLU activation, dropout rate = 0.1, and post-normalization. We use $d_{\mathrm{model}} = 128, d_{\mathrm{ff}} = 256$ for the *DAT* models and $d_{\mathrm{model}} = 144, d_{\mathrm{ff}} = 288$ in the Transformer models to control for parameter count and give the Transformer an advantage in the evaluation. Sinusoidal positional embeddings are used as the positional encoding method. For all models, the total number of attention heads (across self-attention and relational attention) is 8. For the Transformer model, there are only self-attention heads: $n_h^{sa} = 8$ for both the encoder and decoder. For *DAT*, we evaluated two configurations for the composition of head types, one with $n_h^{sa} = n_h^{ra} = 4$ in the encoder and $n_h^{sa} = 8, n_h^{ra} = 0$ in the decoder (i.e., standard Transformer Decoder), and one with $n_h^{sa} = 4 = n_h^{ra} = 4$ in the encoder and $n_h^{sa} = 4 = n_h^{ra} = 4$ in the decoder. The number of cross-attention heads in the decoder is 8 in all cases. No symmetry constraint is made on relational attention. Position-relative symbols are used as the symbol assignment mechanism, and the symbol library is shared across all layers in both the encoder and decoder.

**Training Details.** Each model is trained on each task for 50 epochs. We use the Adam optimizer with $\beta_1 = 0.9, \beta_2 = 0.995$, a learning rate of $6 \times 10^{-4}$, and a batch size of 128. We evaluate and track the per-character accuracy over the course of training. We repeat this process 5 times for each combination of model and task with different random seeds to compute approximate confidence intervals.

FURTHER DISCUSSION, EXPLORATION, & ABLATIONS

Table 2 reports the full set of results obtained for this experiment, including certain configurations omitted from the figure in the main text.

## C.3. Visual Processing (Section 4.3)

EXPERIMENTAL DETAILS

**Dataset details.** In this set of experiments, we use the CIFAR-10 and CIFAR-100 datasets [44] which are datasets of labeled small images. The CIFAR-10 dataset consists of $60,000$ $32 \times 32$ RGB images, evenly split across 10 classes. The CIFAR-100 dataset consists of $60,000$ RGB images of the same size, evenly split across 100 classes.

**Model architectures.** We use a *ViT*-style architecture [41]. RGB images are divided into $4 \times 4$ patches, flattened, linearly embedded into a vector, and fed through an Encoder. We use average pooling followed by an MLP to produce the final prediction. We evaluate 8-layer models with $d_{\mathrm{model}} = d_{\mathrm{ff}} = 384$, GeLU activation, Pre-LayerNormalization, and no

| Task | Model | Parameter Count | # Layers | $d_{\text{model}}$ | Encoder $n_h^{sa}$ | Encoder $n_h^{ra}$ | Decoder $n_h^{sa}$ | Decoder $n_h^{ra}$ | Accuracy |
|---|---|---|---|---|---|---|---|---|---|
| | Transformer | 692K | 2 | 128 | 8 | 0 | 8 | 0 | $62.5 \pm 1.1\%$ |
| | $DAT$ | 783K | 2 | 128 | 4 | 4 | 8 | 0 | $66.5 \pm 1.0\%$ |
| | Transformer | 871K | 2 | 144 | 8 | 0 | 8 | 0 | $64.0 \pm 1.5\%$ |
| algebra__linear_1 | $DAT$ | 1.09M | 3 | 128 | 4 | 4 | 8 | 0 | $68.1 \pm 6.5\%$ |
| | Transformer | 1.3M | 3 | 144 | 8 | 0 | 8 | 0 | $57.0 \pm 2.3\%$ |
| | $DAT$ | 1.43M | 4 | 128 | 4 | 4 | 8 | 0 | $73.1 \pm 1.1\%$ |
| | Transformer | 1.7M | 4 | 144 | 8 | 0 | 8 | 0 | $53.2 \pm 1.1\%$ |
| | Transformer | 692K | 2 | 128 | 8 | 0 | 8 | 0 | $91.1 \pm 0.2\%$ |
| | $DAT$ | 783K | 2 | 128 | 4 | 4 | 8 | 0 | $91.6 \pm 0.6\%$ |
| | Transformer | 871K | 2 | 144 | 8 | 0 | 8 | 0 | $91.4 \pm 0.2\%$ |
| algebra__sequence_next_term | $DAT$ | 1.09M | 3 | 128 | 4 | 4 | 8 | 0 | $97.0 \pm 0.5\%$ |
| | Transformer | 1.3M | 3 | 144 | 8 | 0 | 8 | 0 | $96.1 \pm 0.5\%$ |
| | $DAT$ | 1.43M | 4 | 128 | 4 | 4 | 8 | 0 | $-$ |
| | Transformer | 1.7M | 4 | 144 | 8 | 0 | 8 | 0 | $93.4 \pm 2.0\%$ |
| | Transformer | 692K | 2 | 128 | 8 | 0 | 8 | 0 | $99.9 \pm 0.0\%$ |
| | $DAT$ | 783K | 2 | 128 | 4 | 4 | 8 | 0 | $100.0 \pm 0.0\%$ |
| | Transformer | 871K | 2 | 144 | 8 | 0 | 8 | 0 | $99.9 \pm 0.0\%$ |
| calculus__differentiate | $DAT$ | 1.09M | 3 | 128 | 4 | 4 | 8 | 0 | $-$ |
| | Transformer | 1.3M | 3 | 144 | 8 | 0 | 8 | 0 | $99.9 \pm 0.0\%$ |
| | $DAT$ | 1.43M | 4 | 128 | 4 | 4 | 8 | 0 | $100.0 \pm 0.0\%$ |
| | Transformer | 1.7M | 4 | 144 | 8 | 0 | 8 | 0 | $99.9 \pm 0.0\%$ |
| | Transformer | 692K | 2 | 128 | 8 | 0 | 8 | 0 | $83.3 \pm 0.1\%$ |
| | $DAT$ | 783K | 2 | 128 | 4 | 4 | 8 | 0 | $85.6 \pm 0.0\%$ |
| | Transformer | 871K | 2 | 144 | 8 | 0 | 8 | 0 | $84.5 \pm 0.3\%$ |
| polynomials__add | $DAT$ | 1.09M | 3 | 128 | 4 | 4 | 8 | 0 | $87.8 \pm 0.1\%$ |
| | Transformer | 1.3M | 3 | 144 | 8 | 0 | 8 | 0 | $86.4 \pm 0.3\%$ |
| | $DAT$ | 1.43M | 4 | 128 | 4 | 4 | 8 | 0 | $88.7 \pm 0.0\%$ |
| | Transformer | 1.7M | 4 | 144 | 8 | 0 | 8 | 0 | $87.6 \pm 0.2\%$ |
| | Transformer | 692K | 2 | 128 | 8 | 0 | 8 | 0 | $74.0 \pm 0.7\%$ |
| | $DAT$ | 783K | 2 | 128 | 4 | 4 | 8 | 0 | $77.8 \pm 0.1\%$ |
| | Transformer | 871K | 2 | 144 | 8 | 0 | 8 | 0 | $74.1 \pm 0.6\%$ |
| polynomials__expand | $DAT$ | 1.09M | 3 | 128 | 4 | 4 | 8 | 0 | $-$ |
| | Transformer | 1.3M | 3 | 144 | 8 | 0 | 8 | 0 | $81.0 \pm 1.2\%$ |
| | $DAT$ | 1.43M | 4 | 128 | 4 | 4 | 8 | 0 | $91.4 \pm 0.9\%$ |
| | Transformer | 1.7M | 4 | 144 | 8 | 0 | 8 | 0 | $89.2 \pm 0.5\%$ |

*Table 2.* Full results of mathematical problem-solving experiments. For each task, this table shows the mean test character-level accuracy $\pm$ the standard error of mean for each model configuration.

| Dataset | Model | Parameter Count | # Layers | $d_{\text{model}}$ | $n_h^{sa}$ | $n_h^{ra}$ | Symmetric $r_{ij}$ | Accuracy |
|---|---|---|---|---|---|---|---|---|
| CIFAR-10 | *ViT* | 7.1M | 8 | 384 | 12 | 0 | NA | $86.4 \pm 0.1\%$ |
| | *ViDAT* | 6.0M | 8 | 384 | 6 | 6 | Yes | $89.7 \pm 0.1\%$ |
| | | 6.6M | 8 | 384 | 6 | 6 | No | $89.5 \pm 0.1\%$ |
| CIFAR-100 | *ViT* | 7.2M | 8 | 384 | 12 | 0 | NA | $68.8 \pm 0.2\%$ |
| | *ViDAT* | 6.1M | 8 | 384 | 6 | 6 | Yes | $70.5 \pm 0.1\%$ |
| | | 6.7M | 8 | 384 | 6 | 6 | No | $70.5 \pm 0.1\%$ |

*Table 3.* Ablation over symmetry of $r_{ij}$ in relational attention for image recognition experiments.

| Dataset | Model | Parameter Count | # Layers | $d_{\text{model}}$ | $n_h^{sa}$ | $n_h^{ra}$ | Accuracy |
|---|---|---|---|---|---|---|---|
| CIFAR-10 | *ViT* | 7.1M | 8 | 384 | 12 | 0 | $89.5 \pm 0.1\%$ |
| | *ViDAT* | 6.0M | 8 | 384 | 6 | 6 | $\mathbf{91.7 \pm 0.1}\%$ |
| CIFAR-100 | *ViT* | 7.2M | 8 | 384 | 12 | 0 | $68.2 \pm 0.1\%$ |
| | *ViDAT* | 6.1M | 8 | 384 | 6 | 6 | $\mathbf{70.9 \pm 0.1}\%$ |

*Table 4.* Classification accuracy on CIFAR-10 and CIFAR-100 with AutoAugment data augmentation during training. Each training configuration is repeated 10 times with different random seeds; we report the mean accuracy $\pm$ the standard error of mean. *DAT* continues to outperform the standard Vision Transformer.

dropout. The *ViT* model has $n_h^{sa} = 12$ standard self-attention heads, while the *DAT* model uses both sensory and relational heads, with an even split $n_h^{sa} = n_h^{ra} = 6$. In the main text, we use symmetric relations $r_{ij}$ with the intuition that visual processing involves symmetric attribute-similarity relations. We also carried out experiments with asymmetric relations and discuss the results below. In *DAT*, we use position-relative symbols as the symbol assignment mechanism. Further, we use Grouped Query Attention [67] in *DAT* to reduce the parameter count to account for the added parameters in relational attention.

**Training Details.** We train for 100 epochs. We use the Adam optimizer with a learning rate schedule consisting of a gradual warmup to $10^{-3}$ in the first 5 epochs, followed by a cosine rate decay down to $10^{-5}$. We use the hyperparameters $\beta_1 = 0.9, \beta_2 = 0.999$, and weight decay of $5 \cdot 10^{-5}$. We normalize the images channel-wise such that pixels have mean zero and unit standard deviation. In the results reported in Table 1 in the main text, we use random cropping, MixUp [45], and CutMix [46] as data augmentation techniques during training. We also report results using AutoAugment [68] below.

FURTHER DISCUSSION, EXPLORATION, & ABLATIONS

**Effect of symmetry in $r_{ij}$.** In the main text, Table 1 reports *DAT* results with symmetric relations $r_{ij}$ by imposing $W_q^{\text{rel}} = W_k^{\text{rel}}$. Here, we explore the effect of this choice. Table 3 compares *DAT* models with and without the symmetry constraint. We find no significant difference in performance. Though, we note the smaller parameter count in the symmetric variant.

**Interpretability Visualization. ??** depicts a visualization of the relations learned by a *ViDAT* model trained on the CIFAR dataset. The relations $r_{ij}$ in relational attention can be interpreted as applying the source patch $i$ as a *filter*, comparing it against each patch $j$. The different components of $r_{ij}$ can be seen as analogous to the channels in a convolution operation. We see that some relations in the *ViDAT* model appear to capture a human-interpretable notion of visual similarity across patches.

**Alternative data augmentation.** In the main text, we use random cropping, MixUp, and CutMix data augmentation during training. Here, we report results on an alternative data augmentation technique: AutoAugment [68]. AutoAugment is an optimized set of data augmentation policies, found through a data-dependent automatic search procedure. At each mini-batch, a random sub-policy is chosen which consists of image processing operations such as translation, rotation, or shearing. Table 4 reports results using this data augmentation procedure. We continue to find that *ViDAT* outperforms the standard *ViT* model.

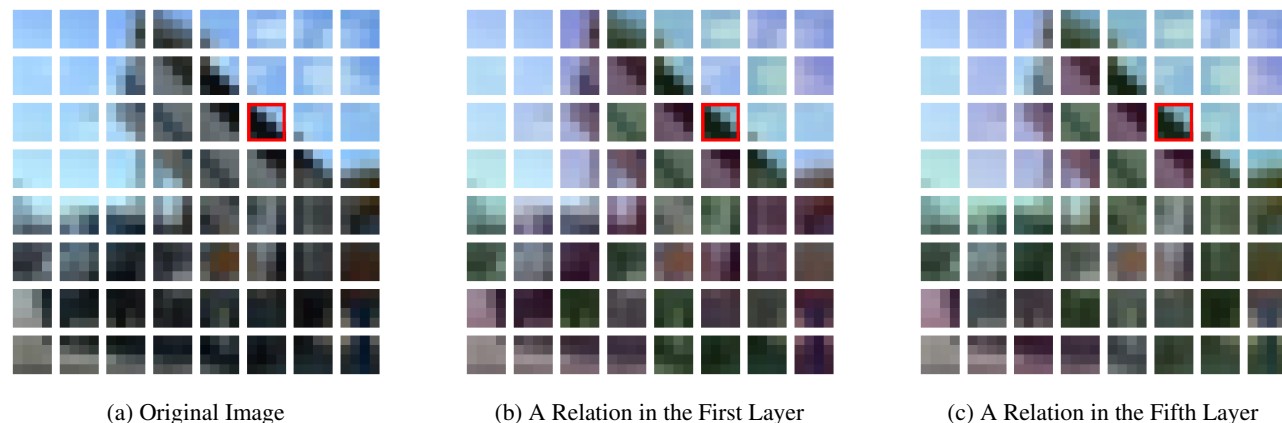

(a) Original Image        (b) A Relation in the First Layer        (c) A Relation in the Fifth Layer

*Figure 9.* A visualization of the relations $\boldsymbol{r}_{ij}[\ell]$ learned by an 8-layer 6M-parameter *ViDAT* model trained on CIFAR. The left panel depicts the original image input. The middle and right panels depict the first relation in the relation vector $\boldsymbol{r}_{ij}$ between the source patch $i$ (red outline) and every other patch $j$. The relation activation is normalized with the tanh function for visualization. The normalized value of the relation is represented by a tint in each patch $j$—a green hue for large positive activations and a red tint for negative activations.

### C.4. Language Modeling (Section 4.4)

Experimental details

**Dataset details.** The FineWeb-Edu [50] dataset is a curated dataset of text data. It is generated by filtering the large-scale FineWeb dataset for LLM pre-training [69] using an educational quality classifier trained on annotations generated by Llama3-70B-instruct. FineWeb-Edu has been shown to outperform FineWeb on several benchmarks, demonstrating the importance of data *quality*. We train our language models on a random subset of 10 billion tokens of FineWeb-Edu.

**Model Architectures.** We use a Decoder-only architecture, with causal attention for autoregressive language modeling. We vary model size to explore the scaling properties of *DAT* with respect to both model size and data size, comparing to the scaling properties of standard Transformers. Our architectural hyperparameters follow common choices at different model scales, based on scaling analyses performed for Transformers [69]. We explore 3 model scales: 350M ($d_{\mathrm{model}} = 1024, n_h = 16, L = 24$), 750M ($d_{\mathrm{model}} = 1536, n_h = 24, L = 24$), and 1.3B ($d_{\mathrm{model}} = 2048, n_h = 32, L = 24$) parameters. We use $d_{\mathrm{ff}} = 4 \cdot d_{\mathrm{model}}$, GeLU activation, RoPE positional encoding, no bias, no dropout, and Pre-LayerNormalization. We use the GPT2 tokenizer [70]. We use symbolic attention as the symbol assignment mechanism, with the number of symbols in the symbol library scaling with model size: 1024 symbols and 8 heads for the 350M and 750M scale models, and 2048 symbols with 16 heads for the 1.3B scale model. We also increase the relation dimension with model size. We don't impose a symmetry constraint, with the intuition that linguistic relations can be asymmetric. We use Grouped Query Attention in the *DAT* models to reduce parameter count to account for the added parameters in relational attention, making them smaller overall compared to the Transformer baselines at each parameter scale.

**Training Details.** We train for 10B Tokens, with each batch containing $524,288$ tokens, split into context windows of $1,024$ tokens. We use gradient accumulation to fit micro-batches into memory. We use the AdamW optimizer with a maximum learning rate of $6 \times 10^{-4}$ and minimum learning rate of $6 \times 10^{-5}$, first linearly warming up over the first 715 steps, then decaying back down with a cosine schedule. We use $\beta_1 = 0.9, \beta_2 = 0.95$ and a weight decay of $0.1$. We also use gradient clipping to unit norm.

Further Discussion, Exploration, & Ablations

Figure 4 in the main text depicts the scaling properties of a *DAT* language model with respect to model size and data size compared to a standard Transformer. Here, we provide a few additional representations of the results. Table 5 reports the end-of-training validation perplexity of the different models.

Figure 9 depicts training curves for the different model scales. We observe a power law scaling of the validation loss with respect to number of training tokens. This matches the neural scaling laws [34], which suggest that validation loss ought to scale roughly as $d^{-\alpha}$ where $d$ is the amount of training data and the exponent $\alpha$ is a constant that depends on model

architecture, training details, etc.

Table 5. End-of-training validation perplexity in language modeling on FineWeb-Edu dataset.

| Model | Param count | # Tokens | $d_{\mathrm{model}}$ | $n_{\mathrm{layers}}$ | $n_h^{sa}$ | $n_h^{ra}$ | $d_r$ | $n_{kv}^h$ | Perplexity $\downarrow$ |
|---|---|---|---|---|---|---|---|---|---|
| Transformer | 353M | 10B | 1024 | 24 | 16 | - | - | - | 16.94 |
| *DAT* | 343M | 10B | 1024 | 24 | 8 | 8 | 64 | 4 | 16.09 |
| Transformer | 757M | 10B | 1536 | 24 | 24 | - | - | - | 14.65 |
| *DAT* | 734M | 10B | 1536 | 24 | 12 | 12 | 64 | 6 | 14.31 |
| Transformer | 1.31B | 10B | 2048 | 24 | 32 | - | - | - | 13.63 |
| *DAT* | 1.27B | 10B | 2048 | 24 | 16 | 16 | 128 | 8 | 13.43 |

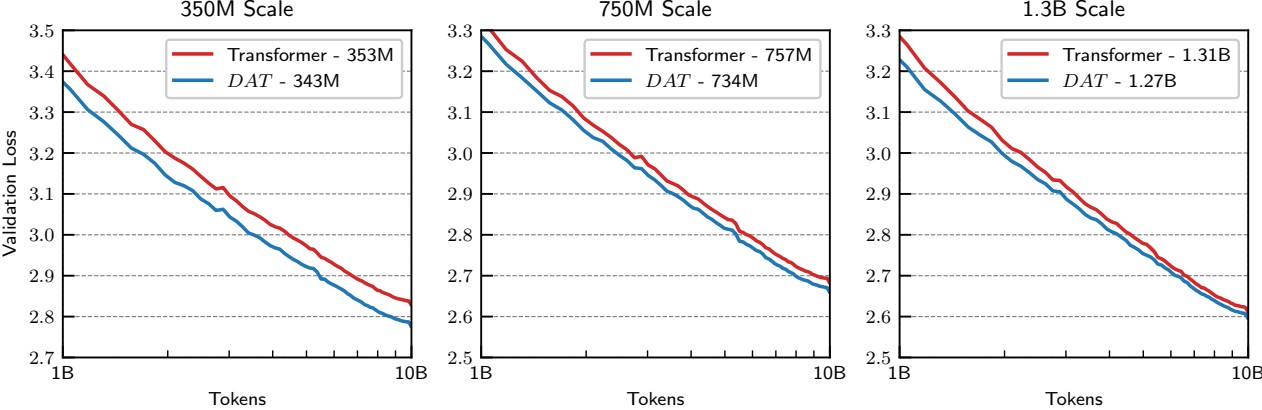

Figure 10. Validation loss on a logarithmic scale to examine data scaling laws. Dual Attention Transformer language models obey similar scaling laws as standard Transformers with respect to the amount of training data, while consistently achieving smaller loss at multiple model scales.

# D. Comparison to Altabaa et al. [22]: Abstractors and Relational Cross-Attention

A closely related work is Altabaa et al. [22], which proposes a Transformer-based module called the "Abstractor" with relational inductive biases. The core operation in the Abstractor is a variant of attention dubbed "relational cross-attention" (RCA). In this section, we will discuss the relation between the Dual Attention Transformer and the Abstractor.

## D.1. Comparison between RA (this work) and RCA [22]

Altabaa et al. [22] propose a variant of attention called relational cross-attention which shares some characteristics with our proposal of what we're calling "relational attention" in this work. In this discussion, we will use the acronyms RCA and RA, respectively to distinguish between the two.

RCA processes a sequence of objects $\boldsymbol{x} = (x_1, \ldots, x_n)$ and produces a sequence of objects $\boldsymbol{x}' = (x_1', \ldots, x_n')$ via the following operation

$$\boldsymbol{x}' \leftarrow \sigma_{\mathrm{rel}}\left(\phi_q(\boldsymbol{x})\phi_k(\boldsymbol{x})^\mathsf{T}\right)\boldsymbol{s},$$
$$\boldsymbol{s} = \mathrm{SymbolRetriever}(\boldsymbol{x})$$

where $\phi_q, \phi_k$ are query and key transformations, and the symbols $\boldsymbol{s}$ take the same role as in this work. $\sigma_{\mathrm{rel}}$ is referred to as a "relation activation". It may be either softmax or an element-wise activation (e.g., tanh, sigmoid, or linear). For the purposes of this discussion, let us consider $\sigma_{\mathrm{rel}} = \mathrm{Softmax}$, which was used in the majority of the experiments in [22].

To facilitate the discussion, let us write RA and RCA side-by-side using a common notation.

<div>

RA (this work)

$$(x_1', \ldots, x_n') \leftarrow \mathrm{RA}(\boldsymbol{x}; S_{\mathrm{lib}}),$$
$$x_i' = \sum_{j=1}^n \alpha_{ij}\left(r(x_i, x_j)\,W_r + s_j\,W_s\right),$$
$$\boldsymbol{\alpha} = \mathrm{Softmax}\left(\phi_q(\boldsymbol{x})\phi_k(\boldsymbol{x})^\mathsf{T}\right),$$
$$r(x, y) = \left(\left\langle \phi_{q,\ell}^{\mathrm{rel}}(x), \phi_{k,\ell}^{\mathrm{rel}}(y)\right\rangle\right)_{\ell \in [d_r]},$$
$$(s_1, \ldots, s_n) = \mathrm{SymbolRetriever}(\boldsymbol{x}; S_{\mathrm{lib}})$$

RCA [22]

$$(x_1', \ldots, x_n') \leftarrow \mathrm{RCA}(\boldsymbol{x}; S_{\mathrm{lib}})$$
$$x_i' = \sum_{j=1}^n \alpha_{ij}\, s_j,$$
$$\boldsymbol{\alpha} = \mathrm{Softmax}\left(\phi_q(\boldsymbol{x})\phi_k(\boldsymbol{x})^\mathsf{T}\right),$$
$$(s_1, \ldots, s_n) = \mathrm{SymbolRetriever}(\boldsymbol{x}; S_{\mathrm{lib}})$$

</div>

RCA can be understood as self-attention, but the values are replaced with symbols (i.e., $\mathrm{Attention}(Q \leftarrow \boldsymbol{x}, K \leftarrow \boldsymbol{x}, V \leftarrow \boldsymbol{s})$). By viewing the attention scores $\alpha_{ij}$ as relations, this has the effect of producing a relation-centric representation. The rationale is that in standard self-attention, the attention scores form a type of relation, but these relations are only used as an intermediate processing step in an information-retrieval operation. The relations encoded in the attention scores are entangled with the object-level features, which have much greater variability. This thinking also motivates the design of RA in the present work.

RCA can be understood as computing a pairwise relation $\langle \phi_q^{\mathrm{attn}}(x_i), \phi_k^{\mathrm{attn}}(x_j)\rangle$ between $x_i$ and each $x_j$ in the context, and retrieving the symbol $s_j$ associated with the object $x_j$ with which the relation is strongest. That is, RCA treats the relations and the attention scores as the same thing. By contrast, the attention operation and computation of relations are separate in RA. The attention component is modeled by one set of query/key maps $\phi_q^{\mathrm{attn}}, \phi_k^{\mathrm{attn}}$ and the relation component is modeled by another set of query/key maps $(\phi_{q,\ell}^{\mathrm{rel}}, \phi_{k,\ell}^{\mathrm{rel}})_{\ell \in [d_r]}$.

The intuitive reason for this choice is that, for many tasks, the optimal "selection criterion" will be different from the task-relevant relation. For example, in a language modeling task, you may want to attend to objects on the basis of proximity and/or syntax while being interested in a relation based on semantics. Similarly, in a vision task, you may want to attend to objects on the basis of proximity, while computing a relation across a certain visual attribute. Thus, the relational attention mechanism proposed in this work offers greater flexibility and expressivity compared to RCA.

In RA, the symbols maintain the role of identifying the source. But they are now explicitly attached to a separately parameterized relation vector.

## D.2. Comparison between *DAT* and the Abstractor

We now briefly discuss the differences in the corresponding model architectures. Altabaa et al. [22] propose an encoder-like module called the Abstractor which consists of essentially replacing self-attention in an Encoder with relational cross-attention. That is, it consists of iteratively performing RCA followed by an MLP. The paper proposes several ways to incorporate this into the broader Transformer architecture. For example, some of the experiments use a Encoder → Abstractor → Decoder architecture to perform a sequence-to-sequence task. Here, the output of a standard Transformer Encoder is fed into an Abstractor, and the Decoder cross-attends to the output of the Abstractor. In another sequence-to-sequence experiment, Altabaa et al. [22] use an architecture where the Decoder cross-attends to both the Encoder and the Abstractor, making use of both sensory and relational information. In particular, the standard encoder and decoder blocks are the same (focusing on sensory information), but an additional module is inserted in between with a relational inductive bias.

By contrast, our approach in this paper is to propose novel encoder and decoder architectures imbued with two distinct types of attention heads, one with an inductive bias for sensory information and the other with an inductive bias for relational information. This has several potential advantages. The first is versatility and generality. The Abstractor architectures that were explored in [22] only explicitly support sequence-to-sequence or discriminative tasks. For example, they do not support autoregressive models like modern decoder-only language models (e.g., of the form we experiment with in Section 4.4). Moreover, even in sequence-to-sequence tasks, Abstractor architectures only support relational processing over the input sequence, but they do not support relational processing over the target sequence (since the decoder does not have RCA). Another potential advantage of *DAT* is simplicity. The Abstractor paper proposes several architectures and configurations for the Encoder/Abstractor/Decoder modules, introducing several hyperparameters that are not trivial to choose. Moreover, it is unclear how to interpret this kind of architecture as the number of layers increases, and the original paper does not experiment with scaling up the number of layers. The final potential advantage is increased expressivity. In *DAT*, the two types of attention heads exist side by side in each layer. This allows relational attention heads to attend to the output of the self-attention heads at the previous layer, and vice-versa. This yields broader representational capacity, and potentially more interesting behavior as we scale the number of layers.

## D.3. How would RCA perform in an *DAT*-style dual head-type architecture?

One question one might ask is: how would an *DAT*-style dual head-type architecture perform if we used Altabaa et al. [22]'s RCA instead of the RA head-type proposed in this work? We carried out a few ablation experiments to answer this question.

Figure 10 compares learning curves on the relational games benchmark between standard *DAT* (with RA-heads) and a version of *DAT* with Altabaa et al. [22]'s RCA heads. We find that the two models perform similarly, with most differences small enough to be within the margin of error. This figure depicts the configuration with asymmetric RA and positional symbols.

Figure 11 depicts the validation loss curves on a small-scale language modeling experiment based on the Tiny Stories dataset [71], comparing standard *DAT* against a version with RCA heads. Here, we find that our relational attention heads yield better-performing models, with the RCA-head variant of *DAT* performing no better than a standard Transformer with a matching total number of heads.

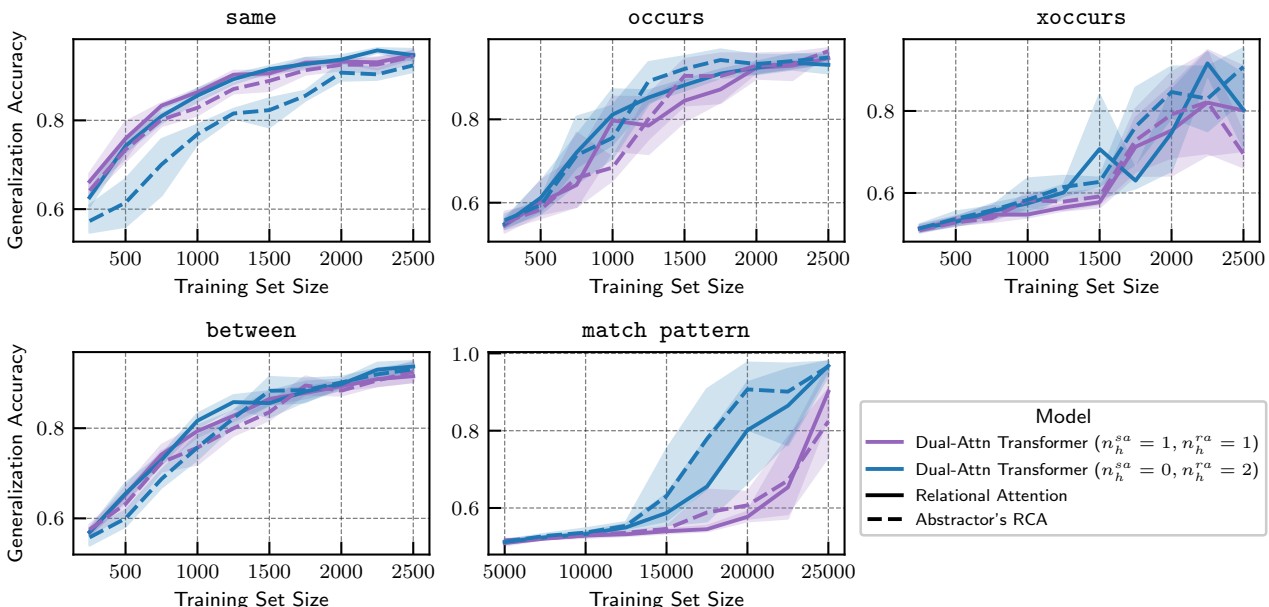

*Figure 11.* Learning curves for *DAT* with RA compared with *DAT* with RCA on the relational games benchmark. The performance is similar, with most differences within the margin of error.

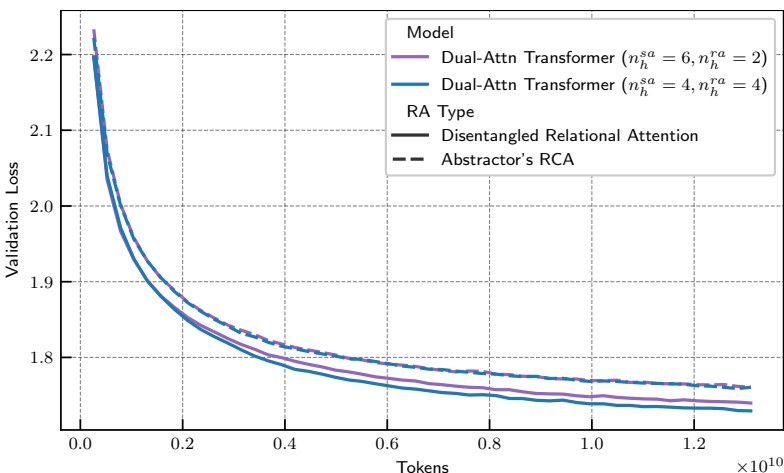

*Figure 12.* Ablation of relational attention type. The solid line depicts the form of relational attention proposed in this work. The dotted line depicts RCA as proposed by Altabaa et al. [22]. We find that our relational attention mechanism performs better, whereas RCA performs no better than a Transformer.

