# OpenReview forum: "Disentangling and Integrating Relational and Sensory Information in Transformer Architectures"
_ICML.cc/2025/Conference — ICML 2025 poster_

### Official Review · Reviewer_pBHH · 2025-03-13

**Overall Recommendation:** 4

**Summary:**

The authors describe a neural architecture (DAT) in which relational information is a first-class object, and via a series of experiments show that this architecture offers genuine empirical benefits.

## update after rebuttal: I have kept my "accept" score for this strong paper. There is no change since my rebuttal responses below, but I am told I need to add this note here as well.

**Claims And Evidence:**

The claims seem solid. The question with this type of paper is often less about the solidity of the claims, however, but more about their significance. To be concrete, why do we need yet another transformer-like architecture? In this case, I think the new architecture meets the bar of significance for publication. First, it can be seen as helping make explicit the type of computation that "classic" transformers are learning implicitly. Second, it appears that for certain tasks DAT actually outperforms transformers. Third, it opens up interesting possibilities for interpretability work, potentially making neural systems more transparent.

**Essential References Not Discussed:**

I think the related work cited is fine. If the authors want to expand a bit, there are two areas where I don't see citations; however, they may not really be essential. One is recent work on how transfomers do seem to represent relations, e.g. "How do Language Models Bind Entities in Context?" (Feng & Steinhardt) and papers that cite it. Another is theoretical ways that have been proposed for representing relations (vector symbolic architecture, etc.)

**Experimental Designs Or Analyses:**

The designs seem solid, and I appreciate that there are both language and vision tasks, which strengthens the overall claim.

**Methods And Evaluation Criteria:**

The paper has a good balance of methods, analyzing the architecture theoretically as well as performing experiments.

**Other Comments Or Suggestions:**

I'm not fond of the term "sensory" here, because it seems actively misleading. There's already a common metaphor for deep networks where people talk about the early layers as doing sensory processing, and the final layers as being analogous to motor neurons.

I'd recommend instead calling this something like "first-order" vs. "relational" information.

**Other Strengths And Weaknesses:**

Figure 8 in the appendix is intriguing, and interpretability work would probably be an entire new paper. I do want to flag one thing about the claim that "Relational attention in DAT language models encodes human-interpretable semantic relations" from the caption: in many cases, classic transformers, too, encode human-interpretable semantic relations. (And here the relation isn't particularly striking: it just seems like it's picking out generally related words?) The fact that one can find a single example of an attention head that is interpretable isn't particularly interesting or useful in itself. I wonder if there's an example of an attention head which finds some kind of relation that is not normally seen in a classic transformer?

**Questions For Authors:**

none, except for the note about confirming my read of the proof, mentioned above.

**Relation To Broader Scientific Literature:**

The question of whether and how transformers learn relationships is definitely central, so this seems squarely in the mainstream. See next section for more detail.

**Theoretical Claims:**

The theorem in Appendix A seems correct, although the Debreu representation theorem was new to me. Once I knew what that was, the result seemed correct. I'd recommend that the authors provide a paragraph outlining the result at a high level, which I read as: selection can be modeled with a preference ordering (with some mild conditions); the Debreu representation theorem allows us to think of computing such a preference ordering as computing a certain continuous function; we then just use standard results saying neural nets can approximate continuous functions. However, I'd note to the authors: if this outline is NOT what the proof is actually saying, then please clarify!

---

> ### Author Rebuttal · Authors · 2025-04-01
>
> Thank you for your detailed and thoughtful review. We appreciate your positive feedback about the significance, methodological soundness, and strength of the empirical evaluation. Below, we hope to address the main concerns you raised in turn.
>
> **D1: Interpretability of Learned Relations**
>
> > Figure 8 in the appendix is intriguing, and interpretability work would probably be an entire new paper. I do want to flag one thing about the claim that "Relational attention in DAT language models encodes human-interpretable semantic relations" from the caption: in many cases, classic transformers, too, encode human-interpretable semantic relations. ... I wonder if there's an example of an attention head which finds some kind of relation that is not normally seen in a classic transformer?
>
> Thank you for your thoughtful comments regarding the interpretability of learned relations. This is indeed an interesting and important question that we’ve begun to explore further. We are glad you found the preliminary exploration in Figure 8 intriguing. As you rightly point out, the relations observed in Figure 8 appear to encode relatively simple semantic similarity between words/tokens, which has also been observed in the attention scores of standard Transformer attention. However, it is important to note that these relations serve a different functional role here: while the attention scores $\alpha_{ij}$ in standard Transformers encode a *selection criterion* controlling *where* information is routed, in *DAT*, the relation vectors $\boldsymbol{r}_{ij}$ serve as the *values* being transmitted between tokens, controlling *what* information is routed (with an independent set of attention scores controlling the selection criterion).
>
> We agree with you that finding a single interpretable relational attention head may not be very revealing on its own, and we would be excited to investigate whether relational heads in *DAT* might encode novel relationships that are not typically seen in classic Transformers. In particular, it would be interesting to understand how these relations form *computational circuits* that are unique to the DAT architecture---that is, characterize specific computational circuits that use relational heads in unique ways to carry out a specific computation, understanding their functional role on a deeper level beyond just the fact that the relations themselves appear human-interpretable.
>
> For now, this interpretability work is outside the scope of this paper, but we intend to explore it further in future work. One initial step we’ve taken is developing an interactive tool (accessible online) that allows users to load pre-trained DAT models and visualize relational representations at various layers on their own inputs. We plan to include a link to this tool in the final, de-anonymized version of the paper.
>
> ---
>
> **D2: Question on proof of Representational Capacity Theorem.**
>
> > The theorem in Appendix A seems correct, although the Debreu representation theorem was new to me ... I'd recommend that the authors provide a paragraph outlining the result at a high level
>
> Thank you for the question and the suggestion to provide further discussion on the Debreu representation theorem. We will add a high-level overview of the Debreu representation theorem and the related literature to improve the clarity and accessibility of the representation result presented in Appendix A
>
> Yes, your interpretation of the result and its proof are correct. The Debreu representation theorem, due to the economics literature, identifies preference relations on a topological (e.g., metric) space with a continuous "utility" function, assuming certain continuity properties of the preference relations with respect to the underlying topology. For us, the key is to extend this idea to a *family* of query-dependent preference relations in order to specify the attention mechanism. That is, each query is associated with an ordered space, and we require continuity of the family of preference relations with respect to both queries and keys, which we formulate as query-continuity and key-continuity, respectively. From there, the result follows by the approximation properties of inner products of MLPs.
>
> **D3: Further discussion of related work**
>
> > If the authors want to expand a bit, there are two areas where I don't see citations; however, they may not really be essential. One is recent work on how transfomers do seem to represent relations, e.g. "How do Language Models Bind Entities in Context?" (Feng & Steinhardt) and papers that cite it. Another is theoretical ways that have been proposed for representing relations (vector symbolic architecture, etc.)
>
> Thank you for these suggestions. We will incorporate them into our discussion of related work in the final version of the paper.
>
> ---
>
> Thank you again for your engagement with our work!

---

> > ### Comment · Reviewer_pBHH · 2025-04-03
> >
> > These sound like useful improvements, and I appreciate the extensive explanations here.

---

### Official Review · Reviewer_yBiU · 2025-03-14

**Overall Recommendation:** 4

**Summary:**

This paper presents the Dual Attention Transformer (DAT), an extension of the Transformer architecture that introduces a relational attention mechanism alongside the standard self-attention mechanism. The key idea is to explicitly represent and process relational information by replacing the standard value aggregation in self-attention with a weighted combination of a relation vector computed for each pair of objects and a symbol vector.

**Claims And Evidence:**

The paper argues that standard Transformers primarily process sensory information and struggle with relational reasoning due to the entanglement of sensory and relational information. The proposed DAT disentangles these components, leading to improved performance. The empirical results largely support these claims.

**Essential References Not Discussed:**

It might be helpful if more key works in graph neural networks that have incorporated relational attention mechanisms were discussed and compared.

**Experimental Designs Or Analyses:**

It might be more convincing if more baseline comparisons with graph-based models and message-passing networks are included, given the conceptual similarities.

**Methods And Evaluation Criteria:**

The authors employ standard benchmarks to assess the effectiveness of DAT, comparing it against standard Transformer models across various tasks.

**Other Comments Or Suggestions:**

The comments and suggestions have been addressed in earlier sections, and no additional ones require emphasis here.

**Other Strengths And Weaknesses:**

The strengths and weaknesses have been addressed in earlier sections, and no additional ones require emphasis here.

**Questions For Authors:**

How does DAT perform on more complex benchmarks beyond the current experimental setup?

**Relation To Broader Scientific Literature:**

The paper presents contribution to the development of relational reasoning in Transformer architectures, with potential implications across multiple domains, including language processing and vision. By introducing an explicit mechanism for processing relational information, the proposed approach highlights the importance of integrating structured reasoning capabilities into deep learning models.

**Theoretical Claims:**

This paper primarily focuses on the empirical aspects.

---

> ### Author Rebuttal · Authors · 2025-03-31
>
> Thank you for your thoughtful review and positive feedback on the overall contribution of our work, especially for highlighting the empirical support for our claims, the contribution to relational reasoning in Transformer architectures, and the importance of structured reasoning in deep learning models. Below, we hope to address the key comments and concerns you raised.
>
> ---
>
> **C1: Comparison with Baselines**
>
> We would like to highlight Appendix C (specifically Section C.1) and Appendix D, where we compare our proposed method to several prior works on relational learning. In particular, we compare our model to PrediNet (Shanahan et al. 2020), CoRelNet (Kerg et al. 2022), and the Abstractor (Altabaa et al. 2024), positioning our work within this ongoing line of research on relational reasoning and extending these prior efforts.
>
> One way to describe the architectures in these prior works is that they incorporate *"subtractive"* inductive biases that constrain the types of representations the model can compute (see [Ref 24] on the "Relational Bottleneck" for more on this). These strict biases enable strong performance on the benchmarks they were designed for (e.g.,  the Relational Games benchmark introduced by Shanahan et al. (2020)), but also constrain the models to a narrow domain of applicability. By contrast, our approach in developing the *Dual Attention Transformer* architecture is *"additive"*, in the sense that it incorporates new explicit relational processing capabilities without constraining existing components of the Transformer architecture, allowing the model to learn to select between the different computational mechanisms available to it based on the task or context, as well as compose them to create flexible and expressive computational circuits.
>
> Despite these differences, we find it valuable to compare *DAT* against those architectures on controlled synthetic benchmarks to explore the trade-offs of strong inductive biases and evaluate *DAT* against alternative approaches to relational learning. This was carried out and discussed in Appendix C.1.
>
> Initially, due to space constraints, we deferred this discussion to the appendix. However, with the additional page allowance, we plan to integrate this more detailed comparison and discussion of related works into the main body of the paper. Please also see our response numbered **A1** to reviewer oPuj, where we discuss a similar question.
>
> ---
>
> **C2: Conceptual similarities to message-passing networks**
>
> We view the aforementioned line of work on relational learning [Ref. 19, 20, 21, 22, 23] to be the most closely related literature to our work. However, we agree that there are some conceptual similarities between our *DAT* architecture and message-passing networks, which is a characteristic shared by Transformer models in general. In particular, like standard Transformers, the *DAT* architecture can be described in the language of the message-passing framework. In message-passing terminology, the messages exchanged in standard Transformer attention encode first-order sensory features of the sender, while in *relational attention*, the messages encode relational features between the sender and the receiver.
>
> We will additionally incorporate a discussion on the conceptual connections between the *DAT* architecture and message-passing networks into the paper.
>
> ---
>
> Thank you again for your review. We hope we were able to address your main concerns.

---

> > ### Comment · Reviewer_yBiU · 2025-04-04
> >
> > Thanks for the response. I've increased my rating.

---

### Official Review · Reviewer_h4pW · 2025-03-14

**Overall Recommendation:** 4

**Summary:**

The authors propose a parameter-efficient variant of the self-attention mechanism in transformers called the Dual-Attention Transformer (DAT), which explicitly routes both sensory information (about individual tokens) and relational information (about relationships between pairs of tokens). The key differences between relational attention and standard attention are that 1) instead of routing a value projection from the query, a vector of relational similarities is constructed (by concatenating the dot products of multiple learnable relation projections between query and key), and 2) A symbol vector (from a learned codebook), retrieved for each key, is added to the corresponding relation vector, and the result is then routed with the usual attention weight.

The authors then incorporate both standard and relational attention heads in a multi-headed attention framework to construct DAT models and demonstrate their effectiveness across various domains - from explicitly relational tasks in the RelationsGame to language modeling, mathematical problem-solving, and image recognition tasks.

**Claims And Evidence:**

The paper makes two primary claims that are well-supported by the evidence:
1. DAT outperforms standard transformers with multi-headed attention - this is demonstrated convincingly across multiple domains and tasks
2. The relational attention mechanism better routes relational information than standard self-attention - this is most clearly shown on explicitly relational tasks from the RelationsGame

**Essential References Not Discussed:**

None of which I am aware

**Experimental Designs Or Analyses:**

The experiments and ablations are thorough, well-reported, and compelling. The learning curves on across most tasks show clear data efficiency advantages of DAT over standard transformers. Furthermore, the additional results in the supplementary material are comprenehsive.

**Methods And Evaluation Criteria:**

The methods and evaluation criteria are appropriate for the demonstrations. The authors evaluate on a diverse set of tasks spanning different domains (visual reasoning, symbolic math problems, language modeling, image classification) and compare against comparable transformer baselines (or relational baselines) while controlling for parameter count.

**Other Comments Or Suggestions:**

* The ICML Style guide requires citations within the text to include authors' last names and year, which this paper doesn't follow
* Line 161 contains an unnecessary sentence fragment "this adds structures..."

**Other Strengths And Weaknesses:**

None of note

**Questions For Authors:**

1. A potential downside of DAT even within the experiments carried out is the possibility that the additional computations required for RelationalAttention outweigh the gains from parameter efficiency. Though differences are likely marginal, was this something you investigated?
2. For very large transformers trained on extensive data, standard self-attention may eventually capture relational information in complex tasks while retaining greater efficacy for other common computations - potentially narrowing the performance gap at increasing scale. While large-scale demonstrations aren't necessary for this paper, do you have intuitions about how the advantages of DAT might scale?

**Relation To Broader Scientific Literature:**

The paper's findings are moderately significant and relevant to current research on improving transformer architectures. The authors position their work appropriately within the literature on relational inductive biases and transformer models.

One potential limitation not fully addressed is whether the performance gains from RelationalAttention would still be significant at very large scales, or if they might be incompatible with optimization tricks for self-attention. However, the consistent improvements across the scales explored are promising and warrant further exploration.

**Theoretical Claims:**

A theoretical claim is made about the class of functions that can be computed by relational attention. A proof is provided in the appendices and was not checked. The claim is not particularly strong and the informal proof in the main text seems sufficient.

---

> ### Author Rebuttal · Authors · 2025-03-31
>
> # Response
>
> Thank you for your detailed and thoughtful review. We appreciate your positive assessment of our work and are encouraged by your recognition of its methodological soundness, strong empirical results, and relevance to the literature on relational inductive biases and Transformer-based architectures. Below, we will aim to respond to the key comments and concerns you raised.
>
> ---
>
> **B1: Computational efficiency**
>
> > A potential downside of DAT even within the experiments carried out is the possibility that the additional computations required for RelationalAttention outweigh the gains from parameter efficiency. Though differences are likely marginal, was this something you investigated?
>
> Thank you for raising this important point. Our experiments were designed to carefully control for model size (i.e., parameter count) when comparing the *DAT* architecture to baselines. In particular, the parameter count is slightly *smaller* for the *DAT* model compared to the baselines. While we did not explicitly measure computational cost in terms of FLOPS, we expect the differences to be marginal.
>
> From a practical standpoint, we believe that the more important factor in computational efficiency is the availability of optimized GPU kernels, such as FlashAttention. This gap could perhaps be bridged given interest from the MLSys community to develop optimized kernels for relational attention, but for now, this would be an obstacle to scaling *DAT*-style architectures in a cost-effective way.
>
> ---
>
> **B2: Scaling to Larger Models**
>
> > One potential limitation not fully addressed is whether the performance gains from RelationalAttention would still be significant at very large scales, or if they might be incompatible with optimization tricks for self-attention. However, the consistent improvements across the scales explored are promising and warrant further exploration.
>
> > While large-scale demonstrations aren't necessary for this paper, do you have intuitions about how the advantages of DAT might scale?
>
> Our intuition, guided by our scaling results up to ~1B-parameter scales, is that the explicit relational processing capabilities of *DAT* will continue to provide significant benefits at even larger scales. Our hypothesis is that relational processing is a key computational capability that is useful in many domains and at several levels of abstraction---explicit support for this in the model architecture can enable more efficient learning and greater generalization capabilities.
>
> Of course, confirming this hypothesis requires empirical validation, which we currently do not have the resources to do at our academic institution. As you rightly mentioned, and as discussed above, scaling further introduces new challenges, and the availability of various optimization tricks becomes an important consideration. We hope future work will explore this further.
>
> ---
>
> Thank you again for your thoughtful review and your engagement with our work.

---

### Official Review · Reviewer_oPuj · 2025-03-15

**Overall Recommendation:** 3

**Summary:**

The authors introduce a modification/extension of the classical attention mechanism they term ‘dual attention’, which not only routes sensory information (as in classic SA) but adds a dedicated pathway to exchange relational information between tokens using its unique attention matrix – allowing for information flow which differs from the sensory one.

**Claims And Evidence:**

All major claims are in my opinion well justified.
The authors do a good job in articulating and introducing their angle on the problem step by step, starting from the well-known Transformer self-attention mechanism. The relational information is explicitly computed (i.e. inductive bias) and added to a symbolic identity, hence justifying the claim that relational information is exchanged. (Although, actual insights into what exactly is exchanged would enhance the paper – currently placed in limited form in the appendix.)

**Essential References Not Discussed:**

None that come to mind – but not extremely up-to-date in this particular area.

**Experimental Designs Or Analyses:**

As previously mentioned, I think the visual experiments could be significantly improved by choosing a task that requires relational modelling in a more obvious way that would be much more intuitive to the reader (e.g. multi-object detection, semantic segmentation, etc.);
It is unclear to me if an instance-based task like classification would benefit from this, as one token might already be enough to determine the correct class label.

Also: Experimental analyses are often performed well but not necessarily contrasted to related methods – this has been deferred to the appendix, but might be better placed in the main paper for visibility and to provide the reader with appropriate context.

**Methods And Evaluation Criteria:**

The authors evaluate their method on four different tasks with corresponding evaluation criteria (acc for vision, perplexity for language, etc.);
The selection of the vision task could in my opinion be significantly improved – Image classification on CIFAR seems rather ill-suited to show the power of relational processing:
- CIFAR is very object-centric and a single-object dataset, hence computation of relations between tokens might be rather straight-forward (around center of image) -- BUT more importantly:
- Image classification: It can be enough to look at one or very few tokens of a CIFAR image and directly tell what the class would be; Hence, this eval seems ill-suited to me.

$\textrightarrow$ There is a variety of other vision tasks where the benefit of relations between parts might be much more intuitive, e.g. multi-object detection, tracking, semantic segmentation, just to name a few.

**Other Comments Or Suggestions:**

Typos:
- L 177 right: to both computational mechanismS (plural)
- L 354 left: it is useful TO consider.. (to missing)

**Other Strengths And Weaknesses:**

**Strengths:**
*Originality & Significance:*
- Clear motivation and step-by-step intro of the method based on a known shortcoming of a missing dedicated modelling mechanism of token-relationships within a Transformer architecture, addressing a known but important gap
- Proposed method applicable to range of modalities due to the choice of a Transformer architecture and the preservation of its generality (in terms of attention)
- Authors demonstrate the applicability via a range of experiments across different tasks/modalities

*Clarity:*
- Explanations well-supported through a good mix of figures, algorithms and formulas
- The paper is well written and easy to read and follow; several details moved to the appendix, but the paper provides a good level of depth to easily follow

---

**Weaknesses:**
- Discussion of & comparison to related works is lacking in parts of the main text: The similarity to [22] is discussed in detail in the appendix, but should be indicated much earlier and in a clearer manner in Sections 2.2 and 2.3; Similarity, almost all experiments exclusively compare default Transformers with DAT – although for some, there are related works available that might even outperform (see appendix Figure 6); Could be discussed in the respective section to be ‘up-front’ with the reader (I don’t expect the method to outperform task-specialist-methods, but a comparison to these works (even if treated in a class of their own) would in my opinion help the reader to better place the proposed method’s strengths
- Experiments on the visual task, i.e. classification on CIFAR, seems rather ill-chosen to support a claim of modelling relations; see previous comments & questions-section
- Details on the ‘subspace’ comparison for relationship computation, i.e. l \in Rd could be extended -- see questions.

**Questions For Authors:**

1. I’d like the authors to provide some more details and insights into the explicit relation computation between the feature maps: The authors state this operation is performed with “$l \in [d_r]$”, and “for each $l \in [d_r]$, the feature maps ..” (l151 f) – producing a relation vector “across different features subspaces”.
$\textrightarrow$ Are these subspaces particularly chosen? And if yes, how?
$\textrightarrow$ How many comparisons are performed between two tokens?
As this is one of the main components of this approach (and differences to [22]), this aspect could be made a lot clearer to the reader.

2. Comments on the visual task: I’d like to know why the authors think that Image classification on CIFAR benefits from the relational modelling. It clearly does, as we can see in the results, but as I’ve mentioned previously:  One token might already be enough to solve the entire task of classifying an image – so this choice doesn’t particularly feel well suited.
$\textrightarrow$  Have the authors visualised what relations are modelled? If not, is this possible and could be included?  Do they represent any ‘expected’/intuitive relations (e.g. parts of the object)?
$\textrightarrow$  The visualisation provided for the language task in Figure 8 (appendix) is quite interesting; so sth similar for the image task would be a good addition;

3. Follow-up:
$\textrightarrow$  Have the authors thought about evaluating their model on an alternative vision task that more intuitively require modelling of relations, e.g. multi-object detection or semantic segmentation?  Why/why now?

*TLDR;* I think the paper presents an interesting approach, although some aspects could be improved (as detailled before); Depending on the response, I’m happy to consider further increasing my score.

**Relation To Broader Scientific Literature:**

Relation to relevant literature can and should be improved – A very important relationship to the work of Altabaa et al. [22] is discussed in the appendix in detail, but the main body of the paper severely lacks in terms of discussing and attributing the similarity;
While indeed different, both method are (in terms of underlying idea, choice of Transformer architecture and modified attention mechanism) very closely related – which should be appropriately indicated already in Sections 2.2 and 2.3.

**Theoretical Claims:**

I have checked the formulas and algorithms, and briefly read through the supporting evidence for Theorem 1 (appendix) – but couldn’t spot any obvious issues;

---

> ### Author Rebuttal · Authors · 2025-03-31
>
> Thank you for your thoughtful and constructive review. We appreciate your positive feedback on the originality, significance, and clarity of our work. We are especially grateful for the time and effort you took to thoroughly engage with our work, reading the appendix and making note of typos. We'd also like to thank you for your specific and constructive feedback, which we believe has helped us improve the paper further. Below, we outline the key concerns you raised and attempt to address them.
>
> ---
>
> **A1: Discussion & Experimental Comparison to Related Work on Relational Architectures (in main body of paper)**
>
> > Experimental analyses are often performed well but not necessarily contrasted to related methods – this has been deferred to the appendix, but might be better placed in the main paper for visibility
>
> > The similarity to [22] is discussed in detail in the appendix, but should be indicated much earlier
>
> We agree that a more detailed discussion of related work in the main text would enhance clarity. Accordingly, we will use the additional page allowance to:
> - Integrate the experimental comparison to previous relational architectures [20,21,22], currently in Appendix C, into the main text.
> - Expand the discussion of Altabaa et al. [22], currently in Appendix D, to Sections 2.2 and 2.3.
>
> ---
>
> **A2: Suitability of CIFAR for Evaluating Relational Processing in Vision**
>
> > CIFAR is very object-centric and a single-object dataset
>
> > I’d like to know why the authors think that Image classification on CIFAR benefits from the relational modelling. It clearly does, as we can see in the results
>
> Although CIFAR datasets contain only one object per image, we believe relational modeling is valuable at multiple levels of abstraction, from local patches to object parts and entire objects. The explicit relational mechanisms in our architecture enable the model to process and reason about visual relationships between object parts and image patches. For instance, this allows the model to detect symmetries or represent visual similarities between object parts across different regions of the image.
>
> > It can be enough to look at one or very few tokens of a CIFAR image and directly tell what the class would be.
>
> In our models, images are divided into small 4x4-pixel patches (64 tokens per image), making it unlikely that a single token would suffice for classification. Since individual tokens represent very small regions at early layers, the models must consider information from several tokens/patches, including the relationships between tokens. At those early layers, the relations visually compare different patches, which can perhaps be thought of as analogous to applying one patch as a "kernel" or "filter" to another patch in the image. At later layers, tokens may come to represent more global higher-level features, and the relations can represent higher-level relations between object parts. Indeed, the improved results we observe for the *DAT* architecture demonstrate the utility of the enhanced relational processing capabilities of our architecture, even for the simple CIFAR benchmark.
>
> That said, we agree that more complex tasks (e.g., multi-object detection, tracking, semantic segmentation) would better showcase our architecture’s capabilities. We will add a discussion on this limitation and potential future directions.
>
> > Have the authors visualised what relations are modelled? If not, is this possible and could be included? Do they represent any ‘expected’/intuitive relations (e.g. parts of the object)?
>
> Following your suggestion, we visualized the relations learned by the *ViDAT* model. We find that some relations do appear to represent intuitive "visual similarity" relations between object parts.
>
> To illustrate this, we provide an example of a visualization of the learned relations on an image of a truck at [Layer 0](https://postimg.cc/PNC0fdLX) and [Layer 4](https://postimg.cc/gXQSL6wY). The patch labeled "source" represents the reference token, and the value annotations indicate sigmoid-normalized relation activations $r_{ij}[\ell]$. The relation activations appear to be high for object parts that are visually similar, especially at earlier layers.
>
> We will add a discussion of these findings in the revised paper.
>
> ---
>
> **A3: Questions**
>
> > Are these subspaces particularly chosen? And if yes, how?
>
> The 'feature subspaces' are not predefined; they are learned via $W_{q,\ell}^{rel}, W_{k, \ell}^{rel}$ during training. These are separate from the $W_{q,h}^{attn}, W_{k,h}^{attn}$ weights, which specify the selection criterion of the attention operation.
>
> > How many comparisons are performed between two tokens?
>
> This is a hyperparameter of the model, denoted $d_r$. For example, in the 1.3B-parameter language model, $d_r = 128$.
>
> We will revise the main text to make these points clearer.
>
> ---
>
> Thank you again for your thoughtful review and your helpful feedback.

---

> > ### Comment · Reviewer_oPuj · 2025-04-05
> >
> > I'd like to thank the authors for their thorough responses to my and the other reviewers' queries.
> >
> > While I do think that the paper provides an interesting approach, I am still not convinced by using *image classification on CIFAR* as a valid way to show the results/validity on vision tasks -- and the visualised relationships on the image provided unfortunately don't really strengthen this aspect. (Compare e.g. various distant sky/background-related patches with higher source-relation against close and distant truck regions)
> >
> > I will therefore keep my rating as is to reflect this aspect.

---

### Decision · Program_Chairs · 2025-05-01

**Decision:**

Accept (poster)

**Comment:**

This paper proposes Dual Attention Transformers, which modifies the standard attention mechanism to not only route sensory information (as in standard SA) but includes a dedicated pathway to exchange relational information between tokens.

This paper received mostly positive reviews, and reviewers agree that the results/claims are well supported. Reviewers laud the quality of the work and in particular the explanation of the proposed mechanism, the diversity of the experiments (both on vision and language), and the comparison to relevant baselines. A few remaining concerns (though not all of them) were well addressed during the rebuttal.  One remaining concern of note is that image classification might not sufficiently test the model's ability to perform relational reasoning token space, even though improved performance is observed. However, given the other experiments and comparisons, this is not a major concern. All in all the reviewers agree this work presents enough of a contribution to be published at ICML and the AC concurs.